# CauseCollab: Causal Unified and Modality-Agnostic Network for Heterogeneous Collaborative Perception

**Weize Li** [* 1]   **Yang Li** [* 1]   **Quan Yuan** [1]   **Xiaoyuan Fu** [1]   **Guiyang Luo** [1]   **Jinglin Li** [1]

## Abstract

Collaborative perception enhances environment understanding through multi-agent information sharing, but its performance in real-world scenarios is constrained by heterogeneous sensor modalities and model architectures. Recent protocol-based two-stage methods alleviate this problem by mapping heterogeneous features into a shared protocol space; however, independently trained modality-specific converters often generate modality-specific pseudo-protocol distributions, leading to semantic inconsistency and error accumulation, which is particularly pronounced in scenarios with large modality discrepancies. To address this issue, we propose CauseCollab, a causal unified and modality-agnostic network. CauseCollab formulates representation learning in the protocol space from a causal perspective, explicitly disentangling semantic factors from modality-specific statistical confounders via causal metric learning. Meanwhile, CauseCollab adopts context-guided Unified Converter for heterogeneous modalities to ensure cross-modal semantic consistency. In addition, integrating new modalities only requires training adapters with minimal parameters. Extensive experiments on the OPV2V and DAIR-V2X datasets demonstrate that CauseCollab achieves state-of-the-art performance, with more significant gains in scenarios involving large modality gaps.

## 1. Introduction

Collaborative perception shares intermediate features among multiple agents through communication, enabling them to complement each other's perceptual blind spots, and is widely regarded as a key technique for improving the robustness of autonomous driving systems (Arnold et al., 2022; Han et al., 2023; Liu et al., 2023). In recent years, extensive research has focused on enhancing the performance of homogeneous collaboration (Liu et al., 2020; Hu et al., 2022; Xu et al., 2022b). However, in real-world deployments, collaborative agents often originate from different manufacturers and platforms, exhibiting substantial differences in sensor types, voxel sizes, and network architectures. It makes intermediate features difficult to fuse and limits the practical effectiveness of collaborative perception.

To address heterogeneity in collaborative perception, prior work has explored feature transformation modules that map heterogeneous modality features from neighboring agents into the local space of the ego agent (Zhao et al., 2023; Yin et al., 2024). Existing approaches can be broadly categorized into two classes. The first class (Xu et al., 2023; Yifan Lu, 2024) learns a dedicated mapping for each pair of heterogeneous modalities to achieve one-stage alignment, but suffers from poor scalability as the training cost grows rapidly with the number of modalities. The second (Luo et al., 2024; Gao et al., 2025) class introduces a unified protocol modality as an intermediate representation space, and trains a converter and a reconstructor for each modality to map heterogeneous features into this protocol space, achieving improved scalability while maintaining strong performance.

Although two-stage heterogeneous adaptation methods based on a protocol space exhibit favorable scalability, their performance is often constrained by error accumulation across stages. When the converters are trained independently for each modality, each modality tends to learn a locally self-consistent mapping between its local representation and the protocol representation. This behavior leads to inconsistent pseudo-protocol distributions in the protocol space. As a result, even after all features are projected into a unified protocol space, significant discrepancies in semantic responses and statistical distributions may persist across modalities, which weakens the effectiveness of local semantic reconstruction.

This issue becomes particularly pronounced when the

---

[*]Equal contribution [1]State Key Laboratory of Networking and Switching Technology, Beijing University of Posts and Telecommunications, Beijing, China. Correspondence to: Quan Yuan <yuanquan@bupt.edu.cn>.

modality gap is large, where pseudo-protocol distributions often introduces noisy responses and spurious activations in background regions. The root cause is that existing methods typically model the protocol space merely as a distribution-alignment intermediary, failing to explicitly characterize which semantic factors should be shared across modalities and which modality-specific statistical confounders should be disentangled.

To address this problem, we propose a unified two-stage framework from a causal perspective. Unlike previous methods that treat the protocol space as a reversible intermediate and emphasize distribution matching, our framework focuses on semantic extraction and modal factor disentanglement in Stage-1, followed by modality-specific local semantic reconstruction in Stage-2.

Specifically, in Stage-1, we adopt a unified converter composed of a semantic context extractor (SCE) and a context-guided dynamic refiner (CGDR) for multi-modality mixed training. By taking global semantics as an anchor to constrain local feature refinement, we reduce background-induced misalignment. Meanwhile, we construct a structural causal model (SCM) (Schölkopf, 2022), depicted in Figure 2, to characterize the relationships among heterogeneous modalities in terms of semantic components. Through a counterfactual intervention strategy based on causal metric learning, we explicitly drive the protocol representations of each modality to approximate the weakly observed protocol modality. This suppresses pseudo-protocol shifts induced by residual modal biases and background noise, thus achieving cross-modal unified semantic alignment. We design the Mask-Guided Intervention via SPD Feature Geometry (MGI-SPD) module to enforce causal interventions. Based on the semantically consistent protocol space, Stage-2 independently trains a local semantic reconstructor for each modality to restore feature semantics to the local modality. When a new modality is introduced, we freeze the backbone of the unified converter and only train lightweight adapters to project the new modality into the semantically consistent protocol space, thus achieving expansion at the cost of minimal parameters.

- We propose CauseCollab, a causal unified and modality-agnostic network for heterogeneous collaborative perception(HCP). It disentangles modality-specific statistical confounders during the conversion, and realizes the learning of a semantically consistent protocol space based on causal metric learning. For intermediate features, we design a module called Mask-Guided Intervention via SPD Feature Geometry.

- We design a unified converter composed of a semantic context extractor and a context-guided dynamic refiner to ensure cross-modal semantic consistency. A lightweight adapter is incorporated to enable the adaptation of new

modalities at the cost of minimal parameters.

- Extensive experiments on the OPV2V and DAIR-V2X datasets demonstrate that CauseCollab achieves state-of-the-art (SOTA) performance among existing heterogeneous collaborative perception methods, with substantial improvements in perception accuracy in scenarios with larger modality gaps.

## 2. Related Work

### 2.1. Collaborative Perception

Collaborative perception allows multiple agents to share and fuse multi-view information, mitigating the inherent view limitations and occlusions of single-agent perception. Intermediate fusion (Li et al., 2021; Xu et al., 2022a) has become the dominant paradigm in collaborative perception, as it preserves rich intermediate semantics while avoiding the high bandwidth requirements and strict synchronization constraints of early fusion. Methods such as Where2comm (Hu et al., 2022) and CodeFilling (Hu et al., 2024) address communication bottlenecks through structured feature compression, while CoAlign (Lu et al., 2023), CoBEVFlow (Wei et al., 2023), and CoDiff (Huang et al., 2025) focus on mitigating feature misalignment caused by localization errors and temporal delays.

Existing collaborative perception methods excel in homogeneous multi-agent settings, yet real-world deployments inevitably require heterogeneous agents. Recently, several works (Xiang et al., 2023; Shao et al., 2024) have attempted to address heterogeneity in collaborative perception. MPDA (Xu et al., 2023) and PnPDA (Luo et al., 2024) employ one-stage and two-stage neighbor-to-ego translators to map features into the ego agent's semantic space. HEAL (Yi-fan Lu, 2024) performs one-to-one mappings via backward alignment, while STAMP (Gao et al., 2025) introduces a protocol space with reversible mappings between local and protocol representations. NegoCollab (Shao et al., 2025) learns a general representation through knowledge distillation. In contrast to prior approaches, our method aims to eliminate pseudo-protocol distributions induced by heterogeneous modalities within the protocol space and to learn a semantically consistent unified protocol representation.

### 2.2. Causal Inference

Causal inference aims to distinguish true causal factors from spurious correlations, characterize causal relationships among variables, and estimate intervention effects that are free from confounding factors. In recent years, causal inference has been introduced into computer vision tasks (Wang et al., 2024; Li et al., 2024) to improve model robustness. CIIM (Yan et al., 2024) explores causally invariant interactions to learn modality-consistent embeddings. CWNet

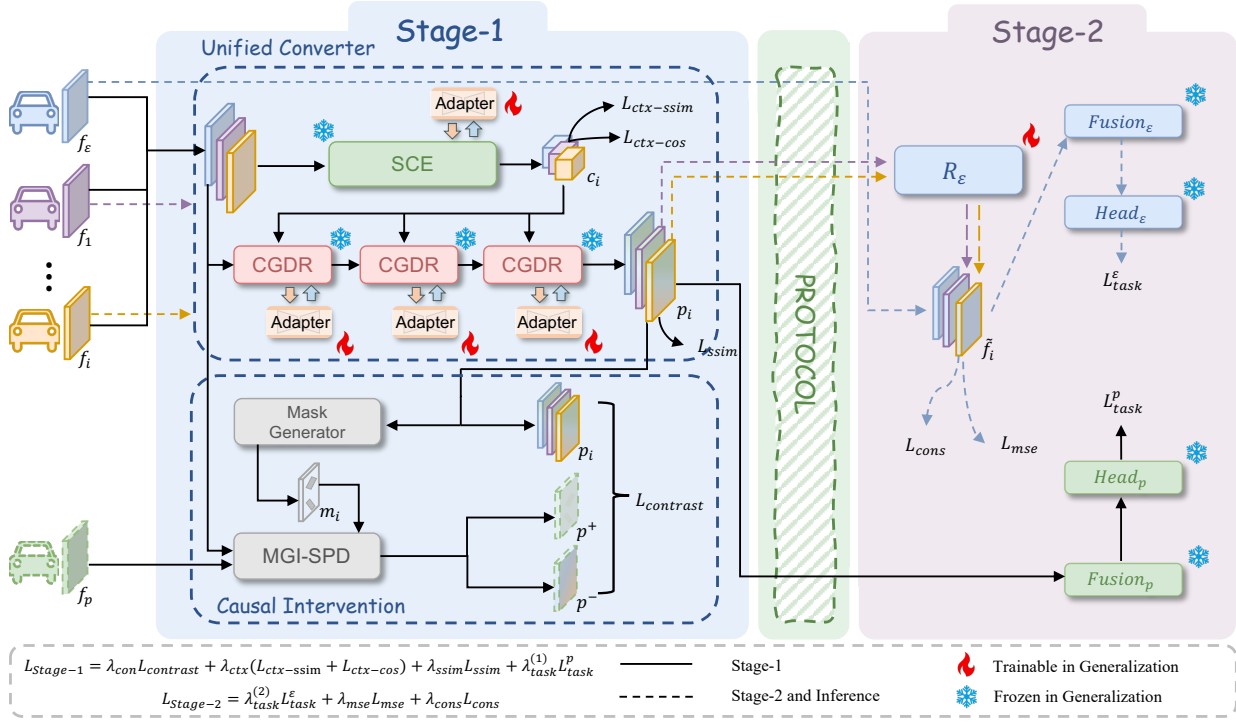

*Figure 1.* The overall architecture of CauseCollab. CauseCollab consists of a modality-agnostic protocol semantics conversion stage via causal modeling (Stage-1) and a modality-specific local semantic reconstruction stage (Stage-2). Unified Converter incorporates SCE and CGDR modules to derive semantically consistent representations.

(Zhang et al., 2025) applies causal reasoning to low-light image enhancement, emphasizing semantic information during the enhancement process. MSFT (Qiao et al., 2025) incorporates causal intervention into time-series forecasting to address confounding effects induced by varying temporal scales. We introduce structural causal models into collaborative perception and design a novel intervention module operating on intermediate features, enabling more robust cross-modal semantic alignment.

## 3. Method

### 3.1. Overview Pipeline

The overall architecture of CauseCollab is illustrated in Figure 1.We focus on the heterogeneous collaborative perception setting, where the ego agent $\varepsilon$ receives observations from multiple collaborative agents $\mathcal{V} = \{1, \ldots, N\}$ during perception. Since collaborative agents may belong to different sensor modalities, the modality set is denoted as $\mathcal{M}$, and directly performing feature fusion suffers from significant modality discrepancies.

For any collaborative agent $i \in \mathcal{V}$, its raw observation is denoted as $x_i^{m_i} \in \mathcal{X}^{m_i}$, where $m_i \in \mathcal{M}$ indicates the sensor modality of agent $i$. The ego agent observation is denoted as $x_\varepsilon^{m_\varepsilon}$. For each modality $m$, we introduce a modality-specific encoder $E^m(\cdot)$ to map raw inputs into feature representations:

$$f_i = E^{m_i}(x_i^{m_i}), \quad f_\varepsilon = E^{m_\varepsilon}(x_\varepsilon^{m_\varepsilon}), \quad (1)$$

where $f_i, f_\varepsilon \in \mathbb{R}^{C \times H \times W}$ denote the modality features of collaborative agents and the ego agent, respectively.

In Stage-1, we introduce a shared protocol space $\mathcal{P}$ and project collaborative agent features into this space to obtain unified semantic representations, where each collaborative feature is transformed into its protocol representation $p_i \in \mathcal{P}$.

In Stage-2, the ego agent employs a reconstructor $R^{m_\varepsilon}(\cdot)$ to map protocol features back to local space $\mathcal{L}$, yielding reconstructed collaborative features $\tilde{f}_i \in \mathcal{L}$. The ego feature $f_\varepsilon$ and all reconstructed collaborative features $\{\tilde{f}_i\}_{i=1}^N$ are then fused by a pyramid fusion module $\Phi_\varepsilon(\cdot)$ to obtain multi-scale fused feature:

$$F_{\text{pyramid}} = \Phi_\varepsilon(f_\varepsilon, \tilde{f}_1, \ldots, \tilde{f}_N). \quad (2)$$

Finally, the fused feature are fed into the task head $H_\varepsilon(\cdot)$ to produce the final detection output:

$$z = H_\varepsilon(F_{\text{pyramid}}). \quad (3)$$

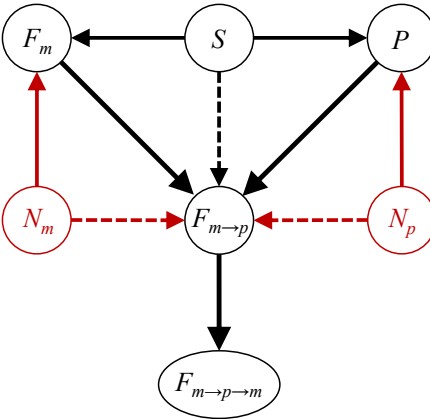

*Figure 2.* Representation-level Structural Causal Model(SCM) for Two-Stage Heterogeneous Collaborative Perception. Black edges denote causal paths, while red edges denote spurious correlation paths. Solid edges represent direct effects, and dashed edges indicate indirect effects on the protocol-space representation.

### 3.2. Stage-1: Protocol Semantic Conversion via Causal Modeling

**Structural Causal Modeling.** To distinguish modality-invariant semantic factors from modality-specific statistical confounders that should be suppressed, we introduce Structural Causal Modeling (SCM) in Stage-1 to explicitly characterize the causal generation process of representations. As illustrated in Figure 2, the variables include the protocol-modality representation $P$, the heterogeneous modality representation $F_m$, the protocol-space heterogeneous representation $F_{m\to p}$, the locally reconstructed representation $F_{m\to p\to m}$, the true semantic factor $S$, the protocol-modality statistical noise $N_p$, and the heterogeneous-modality statistical noise $N_m$.

Without proper constraints, the statistical noises $N_p$ and $N_m$ can affect $F_m$ and $P$ through the encoder, thereby inducing spurious correlation paths in the protocol space that are irrelevant to the true semantics $S$. Consequently, the model may exploit modality-style differences as discriminative cues, leading to modality-specific spurious semantic clusters in $F_{m\to p}$. To obtain a semantically consistent protocol space, we naturally regard $S$ as the causal factor, while treating $N_p$ and $N_m$ as confounders.

Based on the above SCM, the learning objective of Stage-1 can be stated as follows: while maintaining high sensitivity of $F_{m\to p}$ to the semantic factor $S$, we reduce its dependence on modality statistical noises $N_p$ and $N_m$. Since $S$ is not directly observable, we use the protocol-modality representation $P$ as a weak observation during training, and introduce SPD mask-guided intervention together with causal metric learning to achieve the above causal objective.

**Mask-Guided Intervention via SPD Feature Geometry.** After heterogeneous features are transformed into the protocol space, they still retain significant modality-specific statistical shifts, which are mainly reflected in the discrepancies of first-order means and second-order correlation structures of channel distributions (Huang & Belongie, 2017; Li et al., 2017). To address this issue, we design Mask-Guided Intervention via SPD Feature Geometry, and realize the separation of confounders through causal metric learning. Specifically, we regard Bird's-Eye View (BEV) features as a collection of high-dimensional random vectors sampled from various spatial locations, and construct closed-form statistical mappings on the geometry of symmetric positive definite (SPD) covariance matrices (Huang & Gool, 2017); meanwhile, we introduce semantic mask-guided spatially adaptive strength modulation to perform differentiated statistical suppression and statistical injection on foreground and background regions, thereby implementing counterfactual intervention on $N_p$ and $N_m$ without disrupting the feature semantic factor $S$.

Given a protocol feature map $p_i \in \mathbb{R}^{C\times H\times W}$, we reshape it into $X_i \in \mathbb{R}^{C\times N}$ with $N = H \times W$, where each column vector $x_n \in \mathbb{R}^C$ represents the channel feature at the corresponding spatial location. Under this representation, modality style is modeled by the first-order and second-order statistics $(\mu, \Sigma)$ of the feature set $X$.

To characterize the discrepancy in semantic contributions between foreground and background regions, we introduce a semantic mask $m_i \in [0,1]^{1\times N}$ generated by inputting heterogeneous features into a mask generator $g(\cdot)$, and construct non-negative weights $w$ based on this mask to define weighted statistics:

$$\mu = \frac{\sum_{n=1}^{N} w_n x_n}{\sum_{n=1}^{N} w_n}, \tag{4}$$

$$\Sigma = \frac{\sum_{n=1}^{N} w_n (x_n - \mu)(x_n - \mu)^\top}{\sum_{n=1}^{N} w_n} + \varepsilon I, \tag{5}$$

where $\Sigma \in \mathcal{S}_+^C$ lies on the SPD geometry, and $\varepsilon I$ is added to ensure numerical stability.

On the SPD geometry, we adopt a whitening–coloring linear mapping to realize a closed-form transformation from source statistics $(\mu_s, \Sigma_s)$ to target statistics $(\mu_t, \Sigma_t)$:

$$T_{s\to t}(X) = \mu_t + \Sigma_t^{\frac{1}{2}} \Sigma_s^{-\frac{1}{2}}(X - \mu_s), \tag{6}$$

where $\Sigma^{1/2}$ and $\Sigma^{-1/2}$ are computed via eigendecomposition of SPD matrices to ensure numerical stability and preserve positive definiteness. The mapping performs a linear adjustment of the feature mean and correlation structure along the channel dimension only, without altering the spatial structure of the feature map, thereby guaranteeing that the BEV semantics are not compromised.

Under the above formulation, we apply counterfactual intervention to the protocol feature $f_p$ and construct a pair of positive and negative samples for causal metric learning: the demodalized protocol feature $p_i^+$ serves as the positive sample, and the confounded protocol feature $p_i^-$ acts as the negative sample. Specifically, $p_i^+$ is obtained by mapping the channel statistics $(\mu_c, \Sigma_c)$ of $f_p$ to a neutral target distribution $(\mu_0, \Sigma_0)$ via whitening coloring transformation combined with semantic mask-guided spatial interpolation, thereby suppressing modality-specific statistics while preserving the spatial semantic structure; in contrast, after completing the aforementioned suppression, $p_i^-$ is generated by introducing controlled injection of heterogeneous modality statistics $(\mu_h, \Sigma_h)$ through isomorphic statistical mapping and spatial modulation, such that modality-induced statistical perturbations are mixed in while maintaining semantic consistency. Through this intervention construction based on SPD statistical geometry, $p_i^+$ provides a demodalized semantic positive sample, while $p_i^-$ acts as a hard negative sample with controlled modality shifts introduced, jointly driving the unified converter to learn modality-agnostic protocol representations. It is worth noting that MGI-SPD is used only during Stage-1 training to remove modality-specific confounding factors; it is not involved during inference. More details can be found in Appendix B.1.

**Semantic Context Extractor.** To learn representations with stronger semantic consistency in the protocol space, we design a semantic context extractor $G(\cdot)$ for extracting high-level semantic information. This module takes the modality feature $f_i \in \mathbb{R}^{C \times H \times W}$ generated by the heterogeneous modality encoders as input, and outputs the global semantic context feature $c_i \in \mathbb{R}^{C_g \times H_g \times W_g}$.

$G(\cdot)$ adopts a hierarchical structure, and gradually enhances the global semantic modeling capability through two-stage downsampling and semantic extraction. Specifically, we construct two levels of semantic representations:

$$x^{(1)} = \mathcal{B}_1(\mathcal{D}_1(f_i)), \quad x^{(2)} = \mathcal{B}_2(\mathcal{D}_2(x^{(1)})), \quad (7)$$

and fuse the two-level features to obtain the context representation:

$$c_i = \text{Fuse}(x^{(1)}, x^{(2)}). \quad (8)$$

where $\mathcal{D}_1, \mathcal{D}_2$ denote downsampling convolutions, $\mathcal{B}_1, \mathcal{B}_2$ are semantic extraction blocks, and $\text{Fuse}(\cdot)$ is the feature fusion operator. Specifically, $\text{Fuse}(\cdot)$ first aligns the channel dimensions using $1 \times 1$ convolutions, performs spatial alignment via interpolation, and then applies weighted concatenation followed by a depth-wise $3 \times 3$ convolution for spatial mixing. Finally, we apply channel recalibration, implemented by the classical Squeeze-and-Excitation module, and normalization to $c_i$ at the output stage to enhance task-relevant semantic responses and stabilize the feature

distribution, thereby providing reliable global semantic context for subsequent dynamic refinement.

**Context-Guided Dynamic Refiner.** In Stage-1, we introduce the Context-Guided Dynamic Refiner $T(\cdot)$. This module performs layer-wise semantic correction on local modality features under semantic context constraints, thereby generating protocol-space representations with stronger semantic consistency.

Given the local modality feature $f_i \in \mathbb{R}^{C \times H \times W}$ of the $i$-th agent and the semantic context feature $c_i$ extracted by the Semantic Context Extractor (SCE), we first align the context feature in both channel and spatial dimensions to obtain the conditional context representation:

$$\hat{c}_i = U\big(\phi(c_i)\big), \quad (9)$$

where $\phi(\cdot)$ denotes a $1 \times 1$ convolution and $U(\cdot)$ represents linear interpolation. Subsequently, the refiner takes $(f_i, \hat{c}_i)$ as joint input and progressively generates protocol-space features through a stack of context-conditioned transformations:

$$p_i = T(f_i, \hat{c}_i). \quad (10)$$

In practice, the refiner is composed of multiple dynamic refinement blocks. For the $\ell$-th layer, the update of local modality features is conditioned on the aligned context representation, which is formulated as:

$$f_i^{(\ell+1)} = T_\ell\big(f_i^{(\ell)}, \hat{c}_i\big). \quad (11)$$

Within each dynamic refinement block, local features are first modulated by FiLM-style multi-head feature modulation under the guidance of semantic context, and then filtered by a gating function $\text{gate}(\cdot)$, implemented as a $1 \times 1$ convolution followed by SiLU activation, to suppress non-informative regions and noise responses, thereby stabilizing the feature transformation process.

By continuously enforcing semantic context constraints throughout the transformation, the resulting protocol representation $p_i$ not only preserves local discriminative details but also anchors to consistent global semantics, providing reliable inputs for local semantic reconstruction and task prediction in Stage-2.

**Training Strategy.** Stage-1 aims to learn semantically consistent protocol representations: this stage enhances the unified semantic factor $S$ across modalities while suppressing the interference of modality-specific statistical noise $N_p$ and $N_m$, thereby reducing pseudo-protocol shifts and erroneous activations in background regions.

The overall objective function of Stage-1 is defined as fol-

*Table 1.* Performance comparison on the OPV2V dataset using AP@30/AP@50, where the ego agent is $L_{\mathrm{pp4}}$. ALL denotes the scenario with all collaborative agents.

| Method | AP@30 ↑ | | | | AP@50 ↑ | | | |
|---|---|---|---|---|---|---|---|---|
| | $+ C_{\mathrm{Eff}}$ | $+ L_{sd1}$ | $+ C_{\mathrm{Res}}$ | ALL | $+ C_{\mathrm{Eff}}$ | $+ L_{sd1}$ | $+ C_{\mathrm{Res}}$ | ALL |
| MPDA | 0.947 | 0.967 | 0.967 | 0.969 | 0.939 | 0.962 | 0.962 | 0.966 |
| PnPDA | 0.936 | 0.965 | 0.966 | 0.970 | 0.931 | 0.962 | 0.963 | 0.967 |
| HEAL | 0.941 | 0.958 | 0.959 | 0.963 | 0.935 | 0.954 | 0.955 | 0.960 |
| STAMP | 0.945 | 0.967 | 0.968 | 0.972 | 0.940 | 0.964 | 0.965 | 0.969 |
| CauseCollab | **0.952** | **0.972** | **0.973** | **0.978** | **0.944** | **0.968** | **0.969** | **0.975** |

lows:

$$\mathcal{L}_{\text{Stage-1}} = \lambda_{\text{con}}\mathcal{L}_{\text{contrast}} + \lambda_{\text{ctx}}\big(\mathcal{L}_{\text{ctx-ssim}} + \mathcal{L}_{\text{ctx-cos}}\big)$$
$$+ \lambda_{\text{ssim}}\mathcal{L}_{\text{ssim}} + \lambda_{\text{task}}^{(1)}\mathcal{L}_{\text{task}}^{p}, \quad (12)$$

where $\lambda_{\text{con}}$, $\lambda_{\text{ctx}}$, $\lambda_{\text{ssim}}$ and $\lambda_{\text{task}}^{(1)}$ are weighting coefficients.

Among these loss terms, $\mathcal{L}_{\text{contrast}}$ is an InfoNCE-style contrastive loss (van den Oord et al., 2019) for causal metric learning, where $p_i$ is used as the anchor, the demodalized intervention feature $p_i^{+}$ serves as the positive sample, and the confounded intervention feature $p_i^{-}$ serves as the negative sample; $\mathcal{L}_{\text{ctx-ssim}}$ and $\mathcal{L}_{\text{ctx-cos}}$ are semantic context consistency constraints applied between the semantic context feature $c_i$ and the downsampled positive intervention feature $p_i^{+}$, aligning high-dimensional semantics from the perspectives of spatial structure and channel response, respectively; $\mathcal{L}_{\text{ssim}}$ enforces semantic consistency between the protocol-modality feature $f_p$ and the converted protocol representation $p_i$, constraining dynamic refinement to achieve protocol mapping while preserving spatial semantic structure; $\mathcal{L}_{\text{task}}^{p}$ is the protocol-head detection loss, implemented as the focal loss plus the smooth $\mathcal{L}_1$ loss, ensuring protocol representations have both semantic consistency and sufficient discriminability for downstream detection tasks.

### 3.3. Stage-2: Local Semantic Reconstruction

After Stage-1, features from different modalities are mapped into a semantically consistent protocol space. Since heterogeneous modalities differ in resolution and representation characteristics, Stage-2 reconstructs protocol back to each modality-specific local space to fit modality-specific heads.

With the unified converter in Stage-1 frozen, we train a modality-specific local reconstructor $R^m(\cdot)$ for each modality $m$ to map features back to the local feature space:

$$\tilde{f}_i = R^m(p_i), \quad (13)$$

where $p_i$ denotes the protocol-space feature produced by Stage-1. The reconstructor is designed following the ConvNeXt (Liu et al., 2022) architecture to ensure stable optimization.

For modality $m$, Stage-2 is optimized with task supervision and consistency constraints. The overall objective is:

$$\mathcal{L}_{\text{Stage-2}} = \lambda_{\text{task}}^{(2)}\mathcal{L}_{\text{task}}^{\epsilon} + \lambda_{\text{mse}}\mathcal{L}_{mse} + \lambda_{\text{cons}}\mathcal{L}_{cons}, \quad (14)$$

where $\lambda_{\text{task}}^{(2)}$, $\lambda_{\text{mse}}$, and $\lambda_{\text{cons}}$ are weighting coefficients. $\mathcal{L}_{\text{task}}^{\epsilon}$ is the modality-specific task loss for downstream task supervision; $\mathcal{L}_{mse}$ is the MSE loss between reconstructed and original local features; $\mathcal{L}_{cons}$ is the consistency loss to stabilize the mapping between protocol and local feature spaces.

### 3.4. Generalization

The initial collaborative agents have learned modality-agnostic causal semantics with consistent representations in the protocol space. For a new modality, we freeze the backbone of the unified converter in Stage-1 and only employ lightweight adapters to align the new modality to the established protocol-consistent space; in Stage-2, we further train a local semantic reconstructor to recover local space.

The lightweight adaptation in Stage-1 is achieved by inserting a small number of bottleneck adapters at key locations in the unified converter, while keeping the backbone parameters frozen. The adapters are placed after the downsampling operations in the SCE and between modules of the CGDR, so that the new modality is progressively aligned to modality-invariant semantics through scale transitions and fine-grained conversions. Each adapter follows a low-rank bottleneck residual design and is initialized with a scaling coefficient $\alpha = 0$, enabling effective compensation for the new modality with negligible parameter overhead while preserving the original protocol semantic consistency.

## 4. Experiments

### 4.1. Settings

**Datasets.** We conduct experiments on OPV2V (Xu et al., 2022c) and DAIR-V2X (Yu et al., 2022). OPV2V is a

*Table 2.* Experiments of different methods on the OPV2V dataset under the setting of larger domain gap, along with the parameters required for new modality adaptation. The notation $A + B$ indicates that the ego agent is configured with modality $A$, while all neighbor agents are configured with modality $B$. Model parameters are measured in megabytes (MB).

| Method | AP@30/AP@50 ↑ | | | | #Parmas (New Modality) |
|---|---|---|---|---|---|
| | $L_{pp4} + C_{Eff}$ | $C_{Eff} + C_{Res}$ | $L_{pp4} + L_{sd2^{new}}$ | $L_{sd2^{new}} + C_{EffB1^{new}}$ | |
| MPDA | 0.967 / 0.961 | 0.711 / 0.597 | 0.949 / 0.943 | 0.941 / 0.937 | 5.00 |
| PnPDA | 0.966 / 0.962 | 0.654 / 0.597 | 0.980 / 0.979 | 0.951 / 0.946 | 4.37 |
| HEAL | 0.960 / 0.956 | 0.627 / 0.550 | 0.980 / 0.978 | 0.936 / 0.932 | 17.10 |
| STAMP | 0.961 / 0.957 | 0.730 / 0.634 | 0.981 / **0.980** | 0.958 / 0.952 | 1.18 |
| CauseCollab | **0.971 / 0.966** | **0.772 / 0.670** | **0.983 / 0.980** | **0.966 / 0.959** | **0.68** |

*Table 3.* Performance comparison of heterogeneous collaboration on real-world datasets DAIR-V2X.

| Method | AP@30/AP@50 ↑ | |
|---|---|---|
| | $L_{pp4} + C_{Eff}$ | $C_{Eff} + L_{pp4}$ |
| MPDA | 0.792 / 0.685 | 0.405 / 0.228 |
| PnPDA | 0.790 / **0.738** | 0.408 / 0.235 |
| HEAL | 0.738 / 0.692 | 0.410 / 0.253 |
| STAMP | 0.782 / 0.731 | 0.394 / 0.249 |
| CauseCollab | **0.795** / 0.733 | **0.647 / 0.464** |

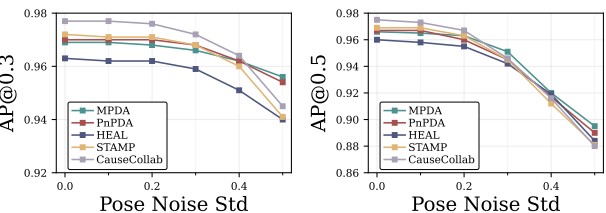

*Figure 3.* Robustness comparison of different methods under pose noise. We evaluate the AP@0.3 and AP@0.5 performance as the pose noise standard deviation increases from 0.0 to 0.5.

simulator-collected multi-agent dataset tailored for heterogeneous collaborative perception. DAIR-V2X is a real-world V2X dataset collected in traffic environments, consisting of both agent-side and roadside LiDAR and camera data.

**Experimental Details.** We adopt heterogeneous feature encoders spanning different sensor types and network architectures. For LiDAR, we use PointPillar (Lang et al., 2019) and SECOND (Yan et al., 2018) as two representative encoders. For cameras, we use an image encoder based on the Lift-Splat-Shoot (LSS) (Philion & Fidler, 2020) framework, which employs ResNet (He et al., 2015) and EfficientNet (Tan & Le, 2019) as the backbone networks. Specifically, we construct four heterogeneous modality configurations, denoted as $L_{pp4}$, $C_{Eff}$, $L_{sd1}$, and $C_{Res}$. These configurations differ in sensor types, voxel size, and backbone design. Detailed settings are provided in Table 6.

We employ a two-stage training scheme. In Stage-1, features extracted by the four heterogeneous encoders are mixed and fed into shared converter. The converter is trained with multi-modal mixed batches to align heterogeneous modality spaces into a unified protocol space. In Stage-2, we train a modality-specific local semantic reconstructor for each modality to recover collaborator features in its local representation space. When integrating a new modality, we freeze the shared converter backbone and only perform lightweight adaptation, together with training the local re-

constructor for the new modality. Notably, during adaptation, the encoders, fusion modules, and classification heads of previously deployed modalities remain frozen and inaccessible, which satisfies practical deployment requirements on privacy preservation and non-modifiability of existing modalities.

### 4.2. Performance Comparison

**Collaboration on OPV2V.** We evaluate our method on the 3D object detection task and compare it with prior approaches. We select two one-stage methods, MPDA (Xu et al., 2023) and HEAL (Yifan Lu, 2024), and two two-stage methods, PnPDA (Luo et al., 2024) and STAMP (Gao et al., 2025). For a fair comparison, we adopt the same pyramid fusion strategy used in prior works (Yifan Lu, 2024; Gao et al., 2025). Table 1 reports the results on OPV2V under a general heterogeneous collaboration setting. We set the ego agent to the $L_{pp4}$ modality and progressively add three heterogeneous-modality neighbors for collaboration. CauseCollab achieves the best performance under both AP@0.3 and AP@0.5 metrics after each heterogeneous modality agent is progressively added.

Furthermore, we evaluate the robustness of the model against localization errors by injecting Gaussian noise with standard deviation $\sigma$ ranging from 0.0 to 0.5 into agent poses, and the experimental results are shown in Figure 3. The results demonstrate that the performance of CauseCol-

*Table 4.* Ablation study: Evaluating the impact of individual components on the performance of CauseCollab. Experiments are conducted under the modality configuration of $L_{\mathrm{pp4}} + C_{\mathrm{Eff}} + L_{\mathrm{sd1}} + C_{\mathrm{Res}}$.

| | AP@30 ↑ | AP@50 ↑ |
|---|---|---|
| CauseCollab | 0.978 | 0.975 |
| -w/o Causal Intervention | 0.969 | 0.965 |
| -w/o Mask Generator | 0.972 | 0.970 |
| -w/o Semantic Context | 0.837 | 0.834 |
| -w/o CGDR | 0.972 | 0.969 |
| -w/o SCE + CGDR | 0.968 | 0.966 |
| -w/o Lightweight Adapters | 0.909 | 0.904 |

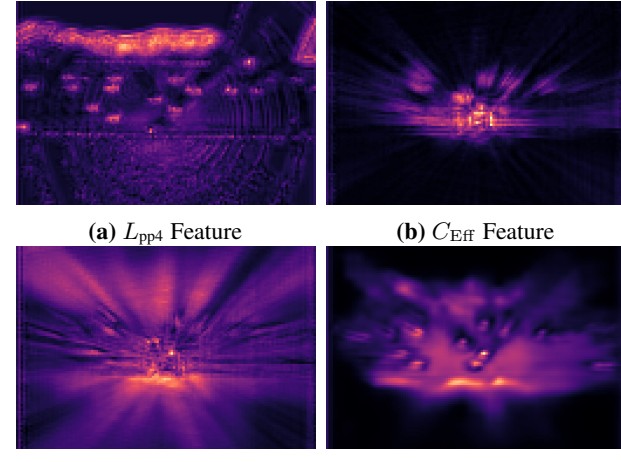

**(a)** $L_{\mathrm{pp4}}$ Feature       **(b)** $C_{\mathrm{Eff}}$ Feature

**(c)** $C_{\mathrm{Eff}}$ Feature in Protocol Space (STAMP)    **(d)** $C_{\mathrm{Eff}}$ Feature in Protocol Space (CauseCollab)

*Figure 4.* Visualization of intermediate features before and after conversion.

lab is better than or comparable to that of other methods at most noise levels.

To validate semantic consistency in the protocol space, we set the ego agent to one modality and neighbor agents to a single different modality, which emulates a larger modality gap for heterogeneous collaboration. We then evaluate the model's robustness in this challenging scenario, with results provided in Table 2. When the ego is $C_{\mathrm{Eff}}$ and the neighbor is $C_{\mathrm{Res}}$, our method improves AP by 4.2% at IoU=0.3 and by 3.6% at IoU=0.5 compared to four other methods. Similarly, when the ego is $L_{\mathrm{pp4}}$ and the neighbor is $C_{\mathrm{Eff}}$, the AP gains are 0.5% and 0.4% under the two thresholds, with improvements even for the higher-performance combination. These results confirm that our method preserves more semantically consistent information during collaboration.

**Collaboration on DAIR-V2X.** We further conduct performance comparisons on the real-world DAIR-V2X dataset. Since DAIR-V2X only provides two agent perspectives for the same scene, we set the Ego agent to the $L_{pp4}$ and $C_{\mathrm{Eff}}$ modalities respectively for collaboration. As shown in Table 3, our method still outperforms all previous methods overall.

**Scalability.** Heterogeneous collaboration methods should be scalable to incorporate novel modalities. Starting from a unified converter trained on the four heterogeneous modalities above, we adapt it to two novel modalities, $L_{\mathrm{sd2}}$ and $C_{\mathrm{EffB1}}$, via fine-tuning. We then use the novel modality as the neighbor agent for collaboration, as shown in Table 2. We observe that, for novel modalities, our method achieves higher average precision with fewer trainable parameters, indicating that the unified converter enables semantic consistency and parameter reuse.

### 4.3. Ablation Study

We conducted a comprehensive ablation study on the OPV2V dataset. In Stage-1, we ablate each component

of the model individually and evaluate the heterogeneous collaboration performance under the general collaboration setting. As shown in Table 4, removing any single component consistently leads to performance degradation.

Specifically, after removing the causal intervention module, the AP metrics at IoU thresholds of 0.3 and 0.5 decrease significantly, respectively, highlighting the critical role of our causal intervention in improving the semantic consistency of the protocol space. In addition, removing the intervention mask also causes performance degradation, verifying the effectiveness of the mask-guided intervention mechanism.

We directly remove the semantic context guidance, which results in a substantial performance drop, demonstrating the necessity of semantic context for the unified converter. Replacing CGDR and SCE + CGDR with a single-tower ConvNeXt of the same number of layers, respectively, leads to additional performance degradation, and this downward trend demonstrates the necessity of the coexistence of SCE and CGDR. Finally, removing the adapter for new modalities causes performance degradation, which demonstrates the value of adapters in incorporating new modalities.

### 4.4. Qualitative Evaluation

To provide more intuitive evidence that CauseCollab learns a semantically consistent protocol space for heterogeneous collaboration, we visualize the intermediate features of LiDAR and camera modalities for the same scene in Figure 4, as well as the protocol space features converted from the camera modality by STAMP and our method, respectively. As shown in Figure 4(a) and (b), a significant semantic gap exists between the two modalities' intermediate features. STAMP introduces semantically irrelevant noise in back-

ground regions (Figure 4(c)), whereas our method achieves cleaner and more effective protocol semantics conversion, better preserving cross-modality semantic consistency (Figure 4(d)).

## 5. Conclusion

In this paper, we propose a causal unified and modality-agnostic network called CauseCollab, to address the issues of semantic inconsistency and modality confounding in the protocol space of heterogeneous collaborative perception. CauseCollab achieves modality-agnostic consistent semantics via causal metric learning, and enables the extension of new modalities to the semantically consistent protocol space by training only lightweight adapters with minimal parameters. Extensive experiments demonstrate the effectiveness of our proposed method.

## Acknowledgements

This work was supported in part by the Natural Science Foundation of China under Grant 62272053 and Grant 62472048, in part by the Beijing Nova Program under Grant 20230484364, in part by Beijing Natural Science Foundation under Grant L242081, and in part by BUPT Excellent Ph.D. Students Foundation.

## Impact Statement

This work advances heterogeneous collaborative perception by addressing semantic inconsistency and modality confounding through a causally unified, modality-agnostic framework. CauseCollab enables seamless integration of new sensing modalities with minimal computational overhead, accelerating the deployment of multi-modal collaborative systems in autonomous driving and intelligent transportation. Potential risks may arise from safety-critical deployment and privacy concerns in multi-agent perception systems. However, our framework mitigates these risks to some extent by improving cross-modal semantic consistency and enabling adaptation without accessing or modifying previously deployed modality-specific encoders, fusion modules, and detection heads.

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

# A. Details of Experiment

## A.1. Dataset

**OPV2V.** OPV2V (Xu et al., 2022c) is a large-scale public dataset for multi-agent collaborative perception research. Generated by the CARLA (Dosovitskiy et al., 2017) simulation platform combined with the OpenCDA (Xu et al., 2021) co-simulation framework, it contains over 70 scenarios with a total of 11,464 frames, covering diverse urban scenarios and various road structures. Each agent is equipped with one 16-channel, one 32-channel, and one 64-channel LiDAR, along with 4 RGB cameras and 4 depth cameras.

**DAIR-V2X.** DAIR-V2X (Yu et al., 2022) is a real-world dataset dedicated to vehicle-infrastructure collaborative perception. The dataset scenarios consist of one vehicle and one Road Side Unit (RSU). The agent is equipped with a 40-channel LiDAR and a 1920×1080 resolution camera, while the RSU is equipped with a 300-channel LiDAR and a camera of the same specification.

## A.2. Training Details

To ensure fair comparisons, we use identical encoder weights across different methods, such that the modality encoders achieve the same detection performance in homogeneous settings. All LiDAR encoders are trained with the perception range of $[-102.4, -51.2, 102.4, 51.2]$, while all camera encoders are trained with $[-51.2, -51.2, 51.2, 51.2]$. All heterogeneous cooperative perception networks are trained and evaluated within $[-51.2, -51.2, 51.2, 51.2]$. Our training is conducted on NVIDIA RTX 4090 GPUs. We use the Adam optimizer with an initial learning rate of $1 \times 10^{-3}$. The unified converter is trained for 10 epochs. Adapting to a new modality only requires 6 epochs to reach aligned semantics. Each modality-specific local semantic reconstructor is trained for 5 epochs. The hyperparameters used during training are shown in Table 5.

*Table 5.* Hyperparameters in experiments.

| Loss Weight | Stage-1 | | | | Stage-2 | | |
|---|---|---|---|---|---|---|---|
| | $\lambda_{\text{con}}$ | $\lambda_{\text{ctx}}$ | $\lambda_{\text{ssim}}$ | $\lambda_{\text{task}}^{(1)}$ | $\lambda_{\text{task}}^{(2)}$ | $\lambda_{\text{mse}}$ | $\lambda_{\text{cons}}$ |
| **Value** | 2.0 | 0.5 | 1.0 | 0.5 | 1.0 | 5.0 | 5.0 |

## A.3. Detailed Configuration of Agents

Our experiments involve eight agent modalities and a protocol modality, whose perception configurations and performance in homogeneous scenarios are shown in Table 6. Here, $L$ denotes the LiDAR modality, and $C$ denotes the Camera modality. In the supplementary experiments, $V_{vn4}$ (Zhou & Tuzel, 2017) and $C_{\text{Res34}}$ (He et al., 2015) are additionally included.

*Table 6.* Configuration and performance of agents

| Agent | Encoder | Voxel Size | Camera-Encoder | AP@0.3 / AP@0.5 ↑ |
|---|---|---|---|---|
| Protocol | PointPillar | 0.4, 0.4, 4 | - | 0.982 / 0.980 |
| $L_{pp4}$ | PointPillar | 0.4, 0.4, 4 | - | 0.972 / 0.968 |
| $C_{\text{Eff}}$ | Lift-Splat-Shoot | - | EfficientNet | 0.790 / 0.706 |
| $L_{sd1}$ | SECOND | 0.1, 0.1, 0.1 | - | 0.975 / 0.971 |
| $C_{\text{Res}}$ | Lift-Splat-Shoot | - | ResNet101(Output Channel = 512) | 0.729 / 0.625 |
| $L_{sd2}$ | SECOND | 0.2, 0.2, 0.2 | - | 0.973 / 0.968 |
| $C_{\text{EffB1}}$ | Lift-Splat-Shoot | - | EfficientNetB1 | 0.652 / 0.579 |
| $L_{vn4}$ | VoxelNet | 0.4, 0.4, 0.4 | - | 0.960 / 0.958 |
| $C_{\text{Res34}}$ | Lift-Splat-Shoot | - | ResNet34(Output Channel = 128) | 0.619 / 0.546 |

# B. Implementation Details

## B.1. Intervention via SPD Feature Geometry

All operations are performed on the reshaped protocol feature

$$p \in \mathbb{R}^{C \times H \times W} \rightarrow X \in \mathbb{R}^{C \times N}, \quad N = H \cdot W,$$

where the $n$-th column $x_n \in \mathbb{R}^C$ denotes the $C$-channel feature vector at spatial position $n$.

We represent modality style using the weighted first- and second-order statistics $(\mu, \Sigma)$ computed from $X$, with $\Sigma \in \mathcal{S}_+^C$. A semantic mask $m \in [0, 1]^{1 \times N}$, produced by a mask generator $g(\cdot)$, is *required* to define spatially varying non-negative weights, thereby modulating foreground and background contributions. The Mask Generator applies the classification head from single-modality training to local features to produce a confidence map, which is filtered by a Gaussian kernel ($k = 5, \sigma = 1.0$), and thresholded at 0.01 to obtain a binary mask.

**Notation and weighted statistics.** Given $X = [x_1, \ldots, x_N]$ and non-negative weights $w = [w_1, \ldots, w_N]$, we compute

$$\mu = \frac{\sum_{n=1}^{N} w_n x_n}{\sum_{n=1}^{N} w_n}, \tag{15}$$

$$\Sigma = \frac{\sum_{n=1}^{N} w_n (x_n - \mu)(x_n - \mu)^\top}{\sum_{n=1}^{N} w_n} + \varepsilon I, \qquad \varepsilon > 0, \tag{16}$$

where $\varepsilon I$ ensures numerical stability. In practice, the weights are parameterized using the semantic mask as

$$w = \eta + \rho\, m, \qquad \eta > 0,\ \rho \geq 0, \tag{17}$$

which guarantees strictly positive weights and allows the mask to control contribution strength.

**SPD and whitening–coloring transform.** For any SPD matrix $\Sigma$, let $\Sigma = U\Lambda U^\top$ be its eigendecomposition and define

$$\Sigma^{\frac{1}{2}} = U\Lambda^{\frac{1}{2}}U^\top, \qquad \Sigma^{-\frac{1}{2}} = U(\Lambda + \varepsilon I)^{-\frac{1}{2}}U^\top,$$

where $\Lambda^{\frac{1}{2}}$ is applied elementwise and $\varepsilon$ is added to the eigenvalues for stability. Given source statistics $(\mu_s, \Sigma_s)$ and target statistics $(\mu_t, \Sigma_t)$, the closed-form whitening–coloring transform applied to $X$ is

$$T_{s \rightarrow t}(X) = \mu_t + \Sigma_t^{\frac{1}{2}} \Sigma_s^{-\frac{1}{2}} (X - \mu_s). \tag{18}$$

This linear mapping preserves the spatial arrangement of columns in $X$ while modifying only channel statistics.

MASK-AWARE CANONICAL SUPPRESSION (MCS)

The objective of MCS is to demodalize a protocol feature by mapping its channel statistics to a neutral canonical distribution $(\mu_0, \Sigma_0)$, while preserving spatial semantics through a mask-guided residual update.

---

**Algorithm 1** Mask-aware Canonical Suppression (MCS)

---

**Require:** $X \in \mathbb{R}^{C \times N}$ (protocol feature), confidence map $M$
**Ensure:** $X_0$ (modality-agnostic feature)
1: $m \leftarrow g(M)$ {semantic mask, $1 \times N$}
2: $w \leftarrow \eta + \rho\, m$ {weights as in (17)}
3: Compute $(\mu_c, \Sigma_c)$ from $X$ using (15)–(16)
4: Choose canonical target $(\mu_0, \Sigma_0)$ (e.g., $(\mathbf{0}, I)$ or batch-average)
5: $\tilde{X} \leftarrow T_{c \rightarrow 0}(X)$ using (18)
6: Compute strength map $\beta(m) \in [0, 1]^{1 \times N}$
7: $X_0 \leftarrow X + (\beta(m) \odot \mathbf{1}_C) \odot (\tilde{X} - X)$
8: **Return:** $X_0$

---

Note: All eigendecompositions apply a small eigenvalue regularizer to ensure $\Sigma^{-\frac{1}{2}}$ is well-defined.

CANONICAL SUPPRESS-THEN-INJECT (STI)

To create hard negative samples and controlled modality perturbations, we first suppress modality bias via MCS and then inject heterogeneous modality statistics $(\mu_s, \Sigma_s)$ computed from a style feature $X_s$. The same semantic mask $m$ (and associated weights $w$) is used so that injection respects the original spatial semantics.

---

**Algorithm 2** Canonical Suppress-Then-Inject (STI)

---

**Require:** $X$ (content protocol feature), $X_s$ (heterogeneous style feature), confidence map $M$
**Ensure:** $X^+$ (modality-agnostic positive), $X^-$ (injected hard negative)
1: $X_0 \leftarrow \text{MCS}(X, M)$
2: $m \leftarrow g(M), \ w \leftarrow \eta + \rho m$
3: Compute $(\mu_0, \Sigma_0)$ from $X_0$ and $(\mu_s, \Sigma_s)$ from $X_s$
4: $\hat{X} \leftarrow T_{0 \to s}(X_0)$
5: Compute injection strength map $\beta^{\text{inj}}(m)$
6: $X^- \leftarrow X_0 + (\beta^{\text{inj}}(m) \odot \mathbf{1}_C) \odot (\hat{X} - X_0)$
7: $X^+ \leftarrow X_0$
8: **Return:** $X^+, X^-$

---

The positive sample $p^+$ is obtained by reshaping $X^+$ back to the original spatial layout and serves as the modality-agnostic anchor for metric learning. The negative sample $p^-$ is the injected feature $X^-$, which is a hard negative created by re-introducing controlled heterogeneous modality statistics while preserving consistent semantics.

## B.2. SCE Architecture

The architecture of the SCE is illustrated in Figure 5, which gradually enhances the global semantic modeling capability through two-step downsampling and ConvNeXt (Liu et al., 2022).

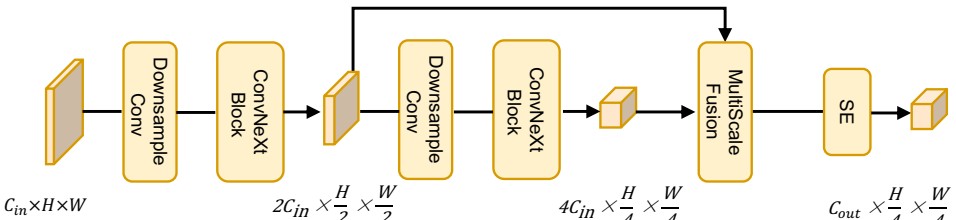

*Figure 5.* Detailed structure of the Semantic Context Extractor

## B.3. CGDR Architecture

The architecture of the CGDR is illustrated in Figure 6. This module takes the local modality feature $f_i \in \mathbb{R}^{C \times H \times W}$ and the context feature $\hat{c}_i \in \mathbb{R}^{C_{ctx} \times H \times W}$ as inputs. After the context feature is processed by a $1 \times 1$ *Project Conv*, it is concatenated with $f_i$ and fed into the *Joint Fusion* module, which consists of a depth-wise convolution and LayerNorm, for information integration.

The fused feature is modulated by the *Context Guide* module (FiLM-style multi-head feature modulation) and recalibrated in channel dimension via the *SE* module (Hu et al., 2019), then filtered by the *Gate* module (1x1 convolution followed by SiLU) to suppress noise. Subsequently, a residual connection with Layer Scale stabilizes the gradient, followed by transformation through an *MLP* and another residual connection with Layer Scale 2 to preserve detailed features. Finally, the output is split into an updated local feature and a context feature for subsequent layers, which balances local discriminability and global semantic consistency.

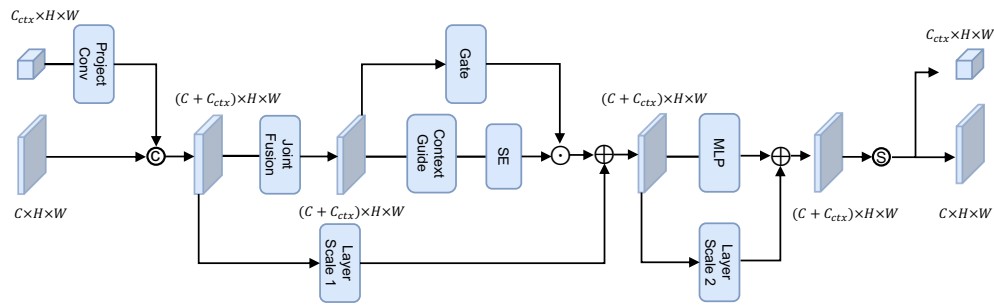

*Figure 6.* Detailed structure of the Context-Guided Dynamic Refiner

# C. More Experiments

## C.1. Comparison between CauseCollab and Late Fusion

Late fusion directly integrates the 3D detection boxes perceived by each agent to enable information sharing. We compare the performance of our method with that of late fusion on the OPV2V dataset under different pose error $\sigma$ values. As shown in Table 7, *ALL* denotes the collaboration of the four modalities. We additionally set two modality configurations for Ego-Neighbor pairs with large modality gaps. Our method significantly outperforms the late fusion method across all pose error levels and collaboration combinations, demonstrating the superiority of our method in real-world application scenarios.

*Table 7.* Performance comparison of CauseCollab and Late Fusion under varying pose error standard deviations

| $\sigma$ | Method | AP@30 | | | AP@50 | | |
|---|---|---|---|---|---|---|---|
| | | ALL | $L_{pp4} + C_{\text{Eff}}$ | $C_{\text{Eff}} + C_{\text{Res}}$ | ALL | $L_{pp4} + C_{\text{Eff}}$ | $C_{\text{Eff}} + C_{\text{Res}}$ |
| 0.0 | Late Fusion | 0.957 | 0.956 | 0.745 | 0.943 | 0.930 | 0.630 |
| | CauseCollab | 0.978 | 0.971 | 0.772 | 0.975 | 0.966 | 0.670 |
| 0.2 | Late Fusion | 0.952 | 0.951 | 0.734 | 0.848 | 0.851 | 0.534 |
| | CauseCollab | 0.976 | 0.970 | 0.762 | 0.968 | 0.960 | 0.625 |
| 0.4 | Late Fusion | 0.830 | 0.849 | 0.624 | 0.563 | 0.632 | 0.332 |
| | CauseCollab | 0.964 | 0.960 | 0.698 | 0.916 | 0.922 | 0.487 |
| 0.6 | Late Fusion | 0.679 | 0.730 | 0.484 | 0.437 | 0.538 | 0.235 |
| | CauseCollab | 0.928 | 0.932 | 0.590 | 0.859 | 0.882 | 0.381 |

## C.2. Analysis of Different Protocol Modalities

The alignment effectiveness of protocol-based heterogeneous collaborative perception depends on the affinity between the protocol modality and heterogeneous modalities. To analyze this effect, we conduct supplementary experiments using either LiDAR ($L_{pp4}$) or Camera ($C_{Eff}$) as the protocol modality. The results are shown in Table 8.

*Table 8.* Performance comparison under different protocol modalities.

| Protocol Modality | ALL | $L_{\text{pp4}} + C_{\text{Eff}}$ | $C_{\text{Eff}} + L_{\text{pp4}}$ |
|---|---|---|---|
| LiDAR (PointPillar) | 0.978 / 0.975 | 0.971 / 0.966 | 0.867 / 0.842 |
| Camera (Lift-Splat-Shoot) | 0.975 / 0.973 | 0.968 / 0.964 | 0.875 / 0.847 |

The results show that CauseCollab achieves comparable collaborative perception performance under both LiDAR-based and Camera-based protocol modalities. This indicates that the proposed causal intervention effectively reduces protocol-modality asymmetry and improves the robustness of the learned protocol space.

## C.3. Comparison between CauseCollab and NegoCollab

NegoCollab (Shao et al., 2025) mitigates the domain gap by negotiating a common representation, whereas our method explicitly disentangles semantic factors from modal entanglement to learn a modality-agnostic protocol space. Experimental results demonstrate that our method outperforms NegoCollab in both general and novel-modality scenarios. **ALL** represents the general multi-agent collaboration scenarios under the same experimental settings as in Table 1, while $L_{sd2^{\text{new}}} + C_{\text{EffB1}^{\text{new}}}$ verifies the performance on the new modality.

*Table 9.* Performance and parameter comparison of CauseCollab and NegoCollab

| Method | ALL | $L_{sd2^{\text{new}}} + C_{\text{EffB1}^{\text{new}}}$ | #Params (MB) |
|---|---|---|---|
| CauseCollab | 0.978/0.975 | 0.966/0.959 | 0.68 |
| NegoCollab | 0.974/0.972 | 0.958/0.953 | 0.98 |

## C.4. Robustness Evaluation on More Novel Modality Adaptation

To further validate the robustness of the CauseCollab protocol space for novel modality adaptation, we introduce two additional novel modalities: $L_{\text{vn4}^{\text{new}}}$ and $C_{\text{Res34}}^{\text{new}}$. Specifically, $L_{\text{vn4}^{\text{new}}}$ denotes a LiDAR modality encoded by VoxelNet (Zhou & Tuzel, 2017), while $C_{\text{Res34}^{\text{new}}}$ denotes a Camera modality based on ResNet34 with a reduced output channel size of 128. Together with $L_{\text{sd2}^{\text{new}}}$ and $C_{\text{EffB1}^{\text{new}}}$, we evaluate a collaborative perception scenario involving four novel modalities: $L_{\text{vn4}^{\text{new}}} + C_{\text{Res34}^{\text{new}}} + L_{\text{sd2}^{\text{new}}} + C_{\text{EffB1}^{\text{new}}}$. The results are reported in Table 10.

*Table 10.* Robustness evaluation under four novel modalities.

| Method | AP@30 | AP@50 |
|---|---|---|
| MPDA | 0.970 | 0.968 |
| PnPDA | 0.968 | 0.965 |
| HEAL | 0.919 | 0.917 |
| STAMP | 0.963 | 0.961 |
| CauseCollab | **0.974** | **0.971** |

## C.5. Additional qualitative experiments

Figure 7 presents additional visualization experiments. To highlight the visual differences, we specifically select $C_{\text{Eff}}$ and $C_{\text{Res}}$ for collaboration due to their larger modality gaps. The visualization intuitively demonstrates the robustness of our method even in such scenarios with larger domain gaps.

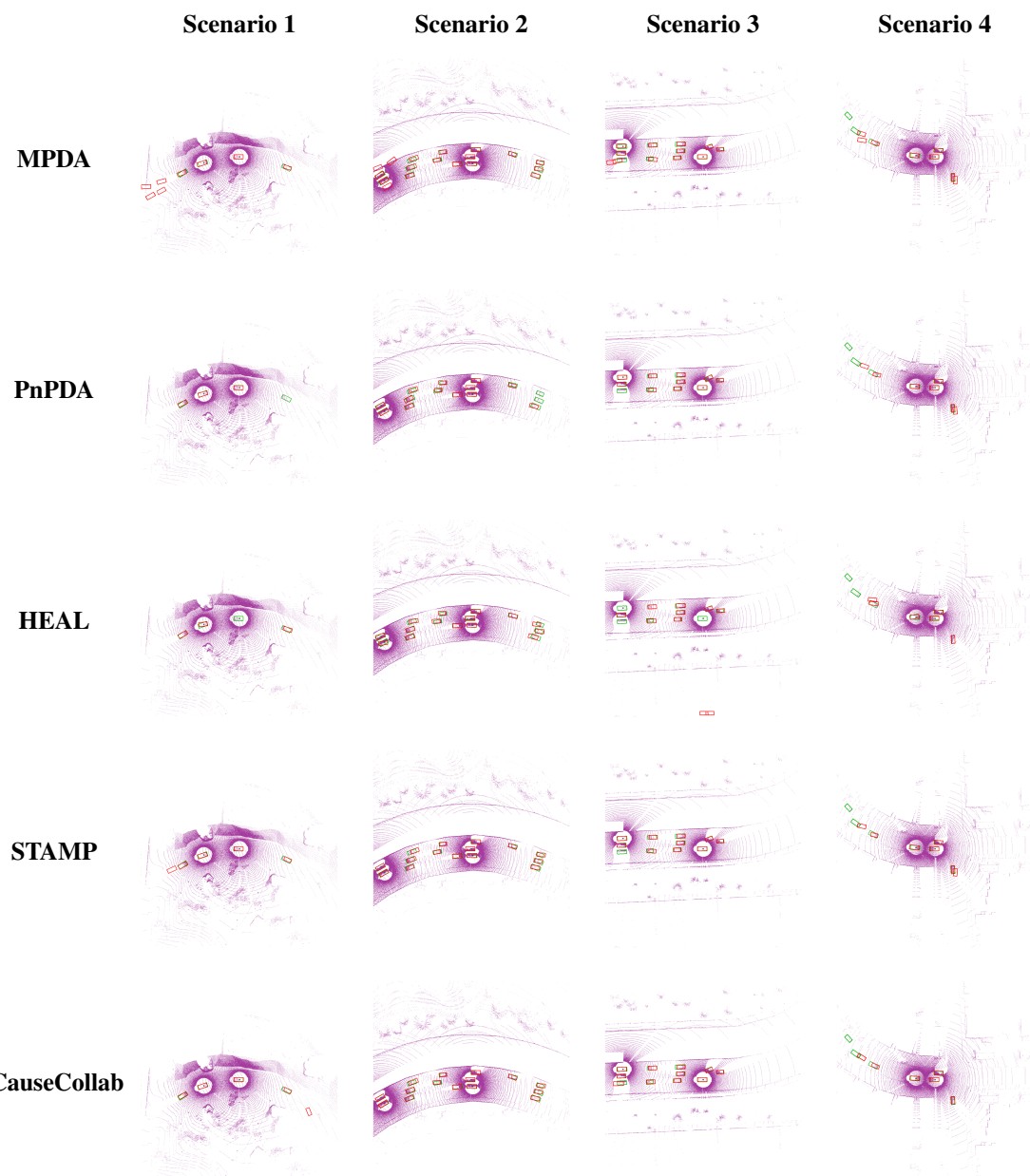

*Figure 7.* Visual comparison of detection results across different methods, using $C_{\text{Eff}}$ and $C_{\text{Res}}$.

