# OpenReview forum: "CauseCollab: Causal Unified and Modality-Agnostic Network for Heterogeneous Collaborative Perception"
_ICML.cc/2026/Conference — ICML 2026 regular_

### Official Review · Reviewer_V1PP · 2026-03-03

**Soundness:** 3
**Presentation:** 3
**Significance:** 2
**Originality:** 2
**Overall Recommendation:** 4
**Confidence:** 2

**Summary:**

Existing HCP methods with protocol spaces don’t explicitly model cross-modal shared information and noisy factors. To tackle this problem, the authors design CauseCollab, which is a two-staged causal unifying framework. At the first stage, the unified converter takes multiple modality features and outputs semantically unified protocol space representations. At the second stage, the local semantic reconstruction takes the unified representations and outputs local modality features for downstream tasks.

**Compliance With Llm Reviewing Policy:**

Affirmed.

**Key Questions For Authors:**

Please see the weaknesses mentioned above.

**Limitations:**

Too many training loss weights may make it hard to reproduce the best performance of CauseCollab.

**Strengths And Weaknesses:**

Strengths:

S1- The paper is well structured, and the figures are informative.

S2- CauseCollab designs multi-modal information networks from different aspects of global and local views.

S3- According to the experiment results, CauseCollab shows great robustness towards the noises compared to the baselines.

Weaknesses:

Although the authors design many training objectives to improve the model performance, the methods introduce too many hyperparameters of the loss weights to be determined. The paper also doesn’t discuss the exact values of these hyperparameters or provide any hyperparameter sensitivity analysis.

---

> ### Author Rebuttal · Authors · 2026-03-31
>
> We thank the reviewer for the recognition of our method and for the valuable suggestions. We sincerely hope the following response addresses your concerns.
>
> For Stage-1 training, we determine the loss weights through grid search. We find that the best AP is relatively sensitive to $\lambda_{\text{con}}$, with the best AP achieved at $\lambda_{\text{con}} = 2.0$. We set $\lambda_{\text{ctx}} = 0.5$, $\lambda_{\text{ssim}} = 1.0$, and $\lambda_{\text{task}}^{(1)} = 5$. For the Stage-2 loss weights, we partially inherit the hyperparameter settings from STAMP (Gao et al., ICLR 2025), while using a smaller $\lambda_{\text{mse}}$. The sensitivity analysis of $\lambda_{\text{con}}$ is reported below:
>
> | $\lambda_{\text{con}}$ | AP\@30 / AP\@50 |
> |:-----------------------------:|:---------------:|
> | 1.5 | 0.975 / 0.972 |
> | 2.0 | 0.978 / 0.975 |
> | 2.5 | 0.977 / 0.974 |
>
> A more detailed hyperparameter sensitivity analysis will be included in the final version.

---

> > ### Author Rebuttal · Reviewer_V1PP · 2026-04-01
> >
> > My concern is fully addressed. Since the hyperparameter setting is crucial for reproducibility, the corresponding details should be included in the revision. I'm going to raise my score.

---

> > > ### Author Response · Authors · 2026-04-01
> > >
> > > We sincerely appreciate the reviewers' thoughtful comments and valuable suggestions.

---

### Official Review · Reviewer_Y5VP · 2026-03-09

**Soundness:** 3
**Presentation:** 3
**Significance:** 3
**Originality:** 3
**Overall Recommendation:** 5
**Confidence:** 3

**Summary:**

This paper studies heterogeneous collaborative perception, where agents may differ in sensing modalities and model architectures. The paper argues that existing protocol-based two-stage methods still suffer from modality-specific pseudo-protocol distributions, which can lead to semantic inconsistency in the shared protocol space, especially under large modality gaps. To address this, the paper proposes CauseCollab, a two-stage framework that combines a context-guided unified converter with a causal-style intervention module to learn a more semantically consistent and modality-agnostic protocol representation. The framework also supports efficient adaptation to new modalities through lightweight adapters. Experiments on OPV2V and DAIR-V2X show state of the art results, with particularly clear gains in large modality-gap settings.

**Compliance With Llm Reviewing Policy:**

Affirmed.

**Final Justification:**

The rebuttal adequately addressed my main concerns, and I now believe the paper is acceptable.

**Key Questions For Authors:**

1. Could the authors clarify in what sense the proposed intervention should be interpreted as causal, rather than feature-level regularization? A clear answer would strengthen the originality claim.

2. In the ablation, removing semantic context seems to hurt more than removing the causal module. Which part of the method contributes most to the final gain? This would affect how I view the main contribution.

3. Could the authors provide more direct evidence that the protocol space is more semantically consistent across modalities, beyond AP and qualitative visualization? A stronger analysis here would make the central claim more convincing.

4. How robust is the learned protocol space under stronger domain shift or substantially different new modalities? A positive answer would strengthen the practical significance.

**Limitations:**

Yes

**Strengths And Weaknesses:**

Strengths

S1. The paper addresses a problem in heterogeneous collaborative perception and clearly motivates why semantic inconsistency remains in protocol-based methods.

S2. The proposed framework is technically coherent, and the unified converter plus lightweight adaptation design is practical for heterogeneous and extensible settings.

S3. The experiments are solid, and the gains are especially clear under large modality gaps, which supports the main empirical claim.

Weaknesses

W1. The causal framing is stronger than the evidence, since the method appears closer to feature-level regularization than clearly validated causal modeling.

W2. The ablation suggests that the main gains may come more from the semantic-context-based converter than from the causal intervention itself.

W3. The claim of improved semantic consistency is supported mainly by downstream AP and visualization, without stronger representation-level evidence.

---

> ### Author Rebuttal · Authors · 2026-03-31
>
> We sincerely thank you for your recognition of our work. Your constructive suggestions are invaluable for improving this work. Below, we address your concerns and present additional experimental results.
>
> > ### Q1
>
> Our causal formulation is established at the representation level. In the protocol space, we explicitly distinguish the semantic factor $S$, which should be preserved, from the modal statistical confounders $N_p$ and $N_m$, which should be suppressed. Accordingly, the objective of Stage-1 is to increase the sensitivity of the protocol representation to $S$ while reducing its dependence on $N_p$ and $N_m$. Unlike conventional feature regularization, which constrains representations only through additional consistency losses, our method introduces an explicit intervention operator.
>
> > ### Q2
>
> To address the reviewer’s concern, we would like to clarify that the ablation on SCE in Table 4 is conducted by removing the contextual correction from SCE to CGDR after end-to-end training, which leads to a substantial performance drop. We further provide results where SCE+CGDR is replaced by a single-tower ConvNeXt with a comparable depth:
>
> | Method | AP@30 | AP@50 |
> |------|-------|-------|
> | CauseCollab | 0.978 | 0.975 |
> | w/o SCE+CGDR | 0.968 | 0.966 |
> | w/o CGDR | 0.972 | 0.969 |
>
> Combined with the ablation results on CGDR, these findings demonstrate the importance of SCE and the necessity of jointly retaining both SCE and CGDR.
>
> > ### Q3
>
> We sample 50 feature instances from the local space, the protocol space of STAMP, and the protocol space of CauseCollab for 2 Lidar and 2 Camera modality, and compute the FID distance and cross-covariance alignment across heterogeneous modality combinations. The results show that the protocol space learned by CauseCollab exhibits stronger semantic consistency.
>
> | Space | Mean FID $\downarrow$ | Same-Ch $\uparrow$ |
> |------|-------|-------|
> | Local | 0.363 | 0.0069 |
> | Protocol (STAMP) | 0.191 | 0.0733 |
> | Protocol (CauseCollab) | 0.177 (-7%) | 0.1063 (+45%) |
>
> where Same-Ch refers to the mean Pearson correlation coefficient between the same channel indices across different modalities.
>
> In addition, the substantial AP gain under the collaborative setting of $C_{\text{Eff}} + L_{\text{pp4}}$ further supports the semantic consistency of our protocol space. The cleaner background in the visualization also suggests that the protocol space is more sensitive to semantic information.
>
> > ### Q4
>
> We thank the reviewer for the insightful question. To further validate the robustness of the CauseCollab protocol space, we introduce additional novel modalities, namely $C_{\text{Res34}}^{\text{new}}$ and $L_{\text{vn4}}^{\text{new}}$. Here, $L_{\text{vn4}}^{\text{new}}$ denotes a LiDAR modality encoded by VoxelNet (Zhou and Tuzel, CVPR 2018)
> , while $C_{\text{Res34}}^{\text{new}}$ denotes a camera modality based on ResNet with a reduced output channel size (128 channels). The collaborative perception results on four novel modalities, $L_{\text{vn4}}^{\text{new}} + C_{\text{Res34}}^{\text{new}} + L_{\text{sd2}}^{\text{new}} + C_{\text{EffB1}}^{\text{new}}$, are reported below:
>
> | Method      | AP\@30 | AP\@50 |
> |:------------|:------:|:------:|
> | CauseCollab | 0.974  | 0.971  |
> | STAMP       | 0.963  | 0.961  |
> | HEAL        | 0.919  | 0.917  |
> | PnPDA       | 0.968  | 0.965  |
> | MPDA        | 0.970  | 0.968  |
>
> These results show that our method exhibits stronger robustness under substantially different novel modalities.
>
> Finally, we sincerely appreciate your constructive comments on our work, and we will include all the aforementioned experiments in the revised manuscript.

---

> > ### Author Rebuttal · Reviewer_Y5VP · 2026-04-03
> >
> > The rebuttal adequately addressed my main concerns, and I now believe the paper is acceptable.

---

> > > ### Author Response · Authors · 2026-04-03
> > >
> > > We sincerely appreciate the thoughtful reviews and helpful suggestions that have strengthened our work.

---

### Official Review · Reviewer_SGUs · 2026-03-12

**Soundness:** 3
**Presentation:** 3
**Significance:** 3
**Originality:** 3
**Overall Recommendation:** 5
**Confidence:** 1

**Summary:**

In this paper the authors propose CauseCollab which uses causal metric learning to address challenges such as semantic inconsistency and modality confounding in collaborative perception.

**Compliance With Llm Reviewing Policy:**

Affirmed.

**Final Justification:**

Given the out-of-scope nature of this work relative to my expertise, I maintain my score which is an educated guess. I have read through the other reviewer comments, but do not feel they have improved my understanding to a degree that I can make further edits or offer further insights.

**Key Questions For Authors:**

1. Is your method potentially applicable to other domains, or is it mainly applicable to collaborative perception?

**Limitations:**

yes

**Strengths And Weaknesses:**

## Strengths
- The paper appears to be well written. Equations and symbols are defined and mostly clear.
- The causal learning angle to approaching the problem of multi-agent heterogenous collaborative perception is well-motivated and represents a potentially valuable conceptual contribution, rather than a purely technical one.
- Results appear strong across multiple datasets, and multiple baselines including both one and two stage methods were used.
- Detailed ablations from each of the components in CauseCollab demonstrate the efficacy of each part (Table 4)
- Inclusion of qualitative results to back up quantitative findings (Fig. 4)

## Suggestions
- There are a lot of symbols and notation to keep track of, some table to refer back to would help significantly in making the paper easier to parse.

---

> ### Author Rebuttal · Authors · 2026-03-31
>
> > ### Q1: Is your method potentially applicable to other domains, or is it mainly applicable to collaborative perception?
>
> We thank the reviewer for the attention and positive feedback. Although our work is developed in the application setting of heterogeneous collaborative perception, its core idea is not limited to this specific task. Instead, it targets a more general problem: how to learn a semantically consistent and scalable shared representation space in the presence of significant modality discrepancies and domain shifts.
>
> More specifically, both the proposed MGI-SPD intervention mechanism and the LoRA-based lightweight adaptation mechanism for novel modalities are inherently general and transferable. We believe that this framework has the potential to be extended to other types of tasks, such as: cross-device representation alignment, where features from different sensors, platforms, or sampling mechanisms need to be coordinated; and federated multi-agent learning, where participating agents have heterogeneous input spaces but are required to collaborate through an intermediate shared representation space.
>
> At the same time, collaborative perception provides a particularly demanding testbed for this problem, as it simultaneously involves heterogeneity, collaboration, modality scalability, and the practical constraint that deployed agents should remain unmodified.

---

> > ### Author Rebuttal · Reviewer_SGUs · 2026-04-03
> >
> > I would like to thank the authors for responding to my question. Since this is not my research area, I will keep my score as I do not have the requisite expertise to make further score updates.

---

> > > ### Author Response · Authors · 2026-04-04
> > >
> > > We sincerely appreciate your thoughtful consideration of our responses and your positive evaluation of our work.

---

### Official Review · Reviewer_RPt6 · 2026-03-13

**Soundness:** 2
**Presentation:** 1
**Significance:** 3
**Originality:** 3
**Overall Recommendation:** 3
**Confidence:** 3

**Summary:**

CauseCollab proposes a two-stage framework for heterogeneous collaborative perception, where multiple autonomous agents — each equipped with a different sensor modality (LiDAR with various voxel sizes, cameras with different backbone architectures) — must share intermediate BEV features for joint 3D object detection. The core problem identified is that existing two-stage protocol-space methods produce *pseudo-protocol distributions*: modality-specific clusters that persist in the shared protocol space because each modality's converter is trained independently, without cross-modal semantic constraints.

The proposed solution has three main components: (1) a Structural Causal Model (SCM) that frames modality-specific noise $(N_m, N_p)$ as confounders of a shared latent semantic factor $S$; (2) Mask-Guided Intervention via SPD Feature Geometry (MGI-SPD), which applies whitening–coloring transforms on SPD covariance matrices to; and (3) a Unified Converter composed of a Semantic Context Extractor (SCE) and a Context-Guided Dynamic Refiner (CGDR), trained jointly on all modalities to enforce semantic consistency. New modalities are integrated via lightweight LoRA-style bottleneck adapters that freeze the shared backbone. Experiments on OPV2V (simulated) and DAIR-V2X (real-world) report state-of-the-art performance.

**Compliance With Llm Reviewing Policy:**

Affirmed.

**Final Justification:**

The authors addressed the majority of my concerns effectively, and I appreciate the additional experiments and implementation details provided. I am updating my score from 2 to 3 to reflect this. However, the paper remains very difficult to follow due to pervasive notation ambiguity and missing implementation details, and one critical concern was not fully addressed, which prevents me from recommending acceptance. I hope a future revision fully incorporating these clarifications will be accepted.

**Key Questions For Authors:**

1. **(Critical — W1)** The SPD whitening–coloring transform suppresses first- and second-order channel statistics $(\mu, \Sigma)$. Can you formally justify why this constitutes a valid causal intervention $\mathrm{do}(N_m = \text{neutral})$ under your proposed SCM? What if modality-specific information is also encoded in higher-order statistics or in spatial activation patterns (e.g., LiDAR point sparsity), which are not addressed by the $(\mu, \Sigma)$ transform? Under what conditions does suppressing $(\mu, \Sigma)$ fully block the causal path from $N_m$ to $F_{m \to p}$?
2. **(Critical — W2)** The protocol modality is specified in Appendix Table 5 as a PointPillar LiDAR encoder (voxel size $0.4 \times 0.4 \times 4$). This means the "neutral" intervention target $(\mu_0, \Sigma_0)$ is implicitly LiDAR-biased. How does performance change if a camera modality (e.g., CRes) is designated as the protocol anchor instead? Does the method maintain modality-agnostic alignment, or do results degrade asymmetrically for LiDAR modalities?
3. **(Critical — W3)** How is the mask generator trained? What is its architecture and supervision signal? Section 3.2.2 states its input is the heterogeneous feature $f_i$, while Algorithm 1 (Appendix B.1) lists the input as a confidence map $M$ — these descriptions conflict. Please clarify and provide sufficient implementation details for reproducibility.
4. **(Important — W4)** What is the concrete implementation of multi-head feature modulation in the CGDR? Specifically: (a) how are modulation parameters derived from $c_i$ (FiLM-style projection, cross-attention, or other); (b) how many heads $H$ are used; (c) how are head outputs aggregated; and (d) what is the architecture of the gating function $g(\cdot)$? Why does $g(\cdot)$ in Section 3.2.4 share the same symbol as the mask generator $g(\cdot)$ in Algorithm 1?
5. **(Important — W5)** How were the loss weighting coefficients $\lambda_{\text{con}}$, $\lambda_{\text{ctx}}$, $\lambda_{\text{ssim}}$, $\lambda^{(1)}_{\text{task}}$ determined, and is performance sensitive to these values? Please also clarify which specific feature pairs each loss term in Equations 12 and 14 operates on.
6. **(Important — W6)** Why is NegoCollab (Shao et al., NeurIPS 2025) absent from the comparison tables despite being discussed in related work as a direct competitor?
7. **(Important — W7)** What is the inference-time overhead of MGI-SPD, particularly the $O(C^3)$ eigendecompositions? How does the latency of CauseCollab compare to STAMP and other baselines?
8. **(Important — W9)** The ablation in Table 4 removes the SCE while retaining a context-free CGDR, but does not compare SCE+CGDR against a single unified end-to-end converter of matched parameter count. Can you provide or describe such a comparison to validate that the two-step decomposition is architecturally necessary?
9. **(Minor — W12)** Why does Equation 2 fuse reconstructed local-space features $\tilde{f}_i$ rather than protocol-space representations $p_i$ directly? Is this because modality-specific detection heads require local-space inputs? If so, this design rationale should be stated explicitly. Is there an ablation comparing these two fusion strategies?

**Limitations:**

The paper includes a brief Impact Statement, but it does not address the negative societal impacts of their work. Given the autonomous driving application, a discussion of the following could better address the limitations:

- Safety: The method is evaluated on average-case AP metrics. No analysis of failure modes, worst-case behavior, or safety-critical edge cases (e.g., missed pedestrian detections due to mask miscalibration) is provided. For life-safety applications, average performance is insufficient.
- Privacy: Sharing intermediate BEV features between vehicles implicitly shares scene-level information (object locations, trajectories). The privacy implications of this feature sharing are not analyzed beyond a brief mention of not sharing model weights.

**Strengths And Weaknesses:**

### Strengths

**S1 — Well-motivated problem framing.** The failure mode of independently trained protocol converters — producing modality-specific pseudo-protocol clusters — is clearly diagnosed and well-illustrated in Figure 4. The causal framing (disentangling semantic factor $S$ from confounders $N_p$, $N_m$) is a coherent conceptual contribution that goes meaningfully beyond prior work's distribution-matching approach.

**S2 — Novel technical combination.** Applying SPD Riemannian geometry (whitening–coloring transforms) to construct counterfactual interventions on BEV feature statistics is a creative and technically interesting design. The mask-guided spatial modulation — applying statistical suppression more aggressively in background regions and more gently in foreground — adds further nuance and is well-motivated.

**S3 — Comprehensive ablation.** Table 4 ablates all five sub-components individually. The SCE ablation is particularly striking: removing the Semantic Context Extractor alone drops AP from 0.978 to 0.837 (−14.1%), the largest single-component drop — larger even than removing the causal intervention module. This is an interesting finding that deserves more discussion.

**S4 — Parameter-efficient generalization.** The adapter-based generalization mechanism (Section 3.4) is well-motivated, clearly described, and supported by strong results: 0.68M parameters for new modality integration versus STAMP's 1.18M and HEAL's 17.1M, with better performance. This is a genuinely practical contribution and Section 3.4 is the best-written section of the paper.

**S5 — Real-world validation and robustness testing.** Including DAIR-V2X alongside simulated OPV2V strengthens credibility. The gains on DAIR-V2X are substantial (e.g., +25.3% AP@0.3 on CEff+Lpp4 over the best baseline). The pose noise robustness experiment (Figure 3) is a thoughtful addition that addresses a realistic deployment concern, and the qualitative feature visualization and discussion in Section 4.4 provide clear, intuitive evidence for the method's effectiveness.

***


### Weaknesses

**W1 (Soundness — Causal justification is shallow).** The SCM in Figure 2 is presented informally, without formal identification assumptions or do-calculus derivations. The core causal claim — that whitening–coloring transforms on channel statistics constitute a valid counterfactual intervention on $N_p$ and $N_m$ — is asserted rather than proven. In structural causal modeling, an intervention $\mathrm{do}(N_m = n)$ requires blocking the causal path from $N_m$ to $F_{m \to p}$; a style-transfer-style normalization only approximates this, to the reviewer's understanding.

Concretely, the intervention assumes modality noise $N_m$ is *fully captured* by first- and second-order channel statistics $(\mu, \Sigma)$. If modality-specific information is also encoded in higher-order statistics or in spatial activation patterns (e.g., LiDAR sparsity patterns at specific spatial locations), the intervention will fail to remove those confounders — and this assumption is never stated or justified. The paper would be significantly stronger if it either (a) proved that the SPD transform satisfies the backdoor or front-door criterion for the proposed causal graph, or (b) more accurately framed the contribution as *causally-motivated metric learning* rather than causal inference proper. Given that the word "causal" appears in the paper's title and is central to its positioning, this is not a minor gap.

Additionally, Figure 2 does not define the distinction between solid and dashed edges — it is unclear whether dashed edges represent bidirected confounding arcs, non-causal associations, or something else. This is non-standard and should be defined. The figure also does not visually indicate which causal paths are severed by the MGI-SPD intervention, which is arguably the most important information the SCM diagram should convey.

**W2 (Soundness — LiDAR-biased protocol anchor).** The protocol modality $P$ — used throughout the method as the semantic anchor for causal intervention and the implicit source of the "neutral" target $(\mu_0, \Sigma_0)$ — is shown in Appendix Table 5 as a PointPillar LiDAR encoder with voxel size $0.4 \times 0.4 \times 4$, a configuration nearly identical to Lpp4. This critical detail is entirely absent from the main paper. In Figure 1, $f_p$ appears without any definition or explanation of its origin. This seems to imply that the protocol modality is Lidar data.

This implication has significant implications that go undiscussed: (1) the "neutral" intervention target $(\mu_0, \Sigma_0)$ is implicitly LiDAR-biased — $\mathrm{do}(N_m = \text{neutral})$ is effectively $\mathrm{do}(N_m = \text{LiDAR-style})$, substituting one modality's noise for another rather than removing the confounder; (2) a real physical protocol agent with a specific sensor must be present during training, which may not always be practical; and (3) the method's claimed modality-agnosticism is asymmetric — camera modalities are aligned toward a LiDAR reference, but not vice versa. No ablation studies what happens when a camera modality is used as the protocol anchor, nor is this design choice discussed.

**W3 (Reproducibility — Mask generator underspecified).** The mask generator is a critical component of MGI-SPD — it determines which spatial regions receive aggressive statistical suppression (background) versus gentle treatment (foreground). A miscalibrated mask could suppress semantic signal in foreground regions or preserve modality noise in background regions, entirely inverting the intended effect. Yet details of the mask generator are underspecified:

- Section 3.2.2 states the mask is generated by "inputting heterogeneous features into a mask generator," while Algorithm 1 (Appendix B.1) lists its input as a confidence map $M$ — these two descriptions are inconsistent.
- Neither the main paper nor the appendix specifies: (1) the architecture of the mask generator; (2) whether it is trained with explicit supervision (e.g., ground-truth object masks, detection head outputs), implicitly end-to-end, or via a heuristic; or (3) what $M$ concretely refers to.

This omission significantly impacts reproducibility and the soundness of the causal intervention claim.

**W4 (Reproducibility — Unified Converter architecture underspecified).** Several key architectural choices in Sections 3.2.3 and 3.2.4 are undefined in the main text and inadequately specified even in the appendix:

- Sections 3.2.3 and 3.2.4 should explicitly direct readers to the relevant appendix sections for architecture details; without such a pointer, $D(\cdot)$, $B(\cdot)$, $\mathrm{Fuse}(\cdot)$, and the modulation mechanism appear undefined.
- The $\mathrm{Fuse}(\cdot)$ operator in Equation 8 is not defined — is it a sum, concatenation, or concatenation followed by a learned projection? This directly affects the output dimensionality of $c_i$.
- "Channel recalibration" applied to $c_i$ is neither named nor cited. If this refers to Squeeze-and-Excitation (Hu et al., 2019), it should be cited explicitly here in Section 3.2.3.
- Section 3.2.4 introduces "multi-head feature modulation under the guidance of semantic context" without any mathematical specification. Is this FiLM-style affine modulation (Perez et al., AAAI 2018), attention-based modulation, multiplicative gating, or something else? How are the modulation parameters derived from $c_i$? How are the $H$ heads aggregated, and what is $H$?
- The "gating function $g(\cdot)$" in Section 3.2.4 is unspecified (sigmoid? softmax? learned threshold?), and crucially, uses the same symbol $g(\cdot)$ as the mask generator in Algorithm 1 (Appendix B.1, line ~657). These are entirely different modules with different inputs, outputs, and purposes. This notation collision should be resolved. In general a lot of notation is introduced in the paper without clear definitions or consistency, which impedes readability and reproducibility.

Collectively, these underspecifications make the method difficult or impossible to reproduce.

**W5 (Reproducibility — Loss functions underspecified).** Equations 12 and 14 list loss terms without specifying which variables they operate on. For example: which feature pairs does $L_{\text{ctx-ssim}}$ compare? Are $L_{\text{ctx-ssim}}$ and $L_{\text{ctx-cos}}$ applied to both $(p_i,p_i^+)$ and $(p_i,p_i^-)$, or to some other pair? What exactly is $L_{\text{ssim}}$ — is it SSIM between refined features and protocol features, and if so which specific pairs? Is $L_{\text{contrast}}$ a standard InfoNCE/NT-Xent formulation or a custom variant? Is the detection task loss a standard regression + classification loss on bounding box coordinates and class labels? If the full loss function is not specified, papers should be cited for each loss term.

Writing each loss as an explicit function of its arguments (e.g., $L_{\text{ssim}}(p_i, f_p)$) would greatly improve clarity and reproducibility. Additionally, the loss weighting coefficients $\lambda_{\text{con}}$, $\lambda_{\text{ctx}}$, $\lambda_{\text{ssim}}$, $\lambda^{(1)}_{\text{task}}$ are never explained — were they set by grid search, manual tuning, or derived from prior knowledge?

**W6 (Presentation — Missing NegoCollab baseline).** NegoCollab (Shao et al., NeurIPS 2025) is discussed in Section 2.1 as a directly relevant approach using knowledge distillation for learning common representations, but is absent from all comparison tables (Tables 1–3). Given its recent publication and direct relevance to the problem, this omission should either be justified or the comparison added.

**W7 (Presentation — No computational cost analysis).** The paper provides no FLOPs, latency, or memory footprint analysis. MGI-SPD requires eigendecompositions of $C \times C$ SPD matrices per sample — an $O(C^3)$ operation that could be a significant inference-time bottleneck for large feature channel counts. At minimum, a runtime comparison with baselines in the appendix is needed for a fair assessment of the method's practical viability in autonomous driving deployments and to support the claim of "efficient generalization" in Section 3.4. Even if the adapters are parameter-efficient, the computational cost or run-time of MGI-SPD could be prohibitive.

**W8 (Presentation — Dataset configuration unclear).** The paper cites OPV2V (Xu et al., 2022c), but the original OPV2V is a simulated with multiple modalities for multiple agents. It is unclear whether each agent has a single modality assignment or accesses multiple sensor types. This should be clarified explicitly in Section 4.1. Similarly, Table 5 presents modality configurations (Lpp4, CEff, Lsd1, CRes) but does not make clear whether these represent different physical sensor types or simply different network architectures applied to the same input data — presumably L is LiDAR and C is camera, but this is never stated.

**W9 (Soundness — SCE+CGDR decomposition not ablated against a unified baseline).** The two-step design of SCE + CGDR is presented as a key architectural contribution, but the ablation in Table 4 only evaluates removing the SCE while retaining a context-free CGDR — this is not equivalent to comparing against a single unified converter $f_i \to p_i$ of matched parameter count. It is unclear whether the explicit context-extraction-then-guidance decomposition is architecturally necessary, or whether a sufficiently expressive end-to-end module would learn the same behavior implicitly. An ablation replacing SCE+CGDR with a single ConvNeXt-based converter of matched parameter count would directly test this. Alternatively, an intuitive argument for why the decomposition helps — supported by visualizations or experiments (e.g., showing that $c_i$ captures different structure than $p_i$) — would address this concern.

**W10 (Presentation — Notation inconsistencies).** Several notation issues impede readability:

- Lines 190–193 (col. 1) introduce $F_m$, $F_{m \to p}$, and $F_{m \to p \to m}$ in the SCM section without clarifying whether these are the same as $f_i$, $p_i$, and $\tilde{f}_i$ introduced in Section 3.1, or distinct variables with different scopes.
- Line 246–247 states that the SCE "takes the modality feature $f_i$ generated by the ego-agent encoder as input." If this is specifically the ego agent's feature, should the notation be $f_\varepsilon$ rather than $f_i$? Does the SCE operate only on the ego encoder, or on all agents' encoders? This is an important functional distinction.
- The use of $P$ throughout Section 3.2.1 is ambiguous — does $P$ refer to the single protocol modality representation (from the anchor agent), or to the space of all $p_i$ representations? The paper uses both $P$ and $p_i$ without clearly distinguishing the anchor modality from per-agent protocol representations.

**W11 (Presentation — $p_i^+$ and $p_i^-$ not written out explicitly).** Lines 223–234 describe the construction of the positive and negative samples in prose but never provide explicit equations for $p_i^+$ and $p_i^-$ as functions of $p_i$, $(\mu_c, \Sigma_c)$, $(\mu_0, \Sigma_0)$, and $(\mu_h, \Sigma_h)$. Writing these out explicitly would considerably clarify the intervention procedure. Furthermore, the values of $(\mu_h, \Sigma_h)$ and $(\mu_0, \Sigma_0)$ are not definitively specified — Algorithm 1 (Appendix B.1) lists $(\mu_0, \Sigma_0)$ as "e.g., $(0, I)$ or batch-average," which was used in the experiments? This ambiguity directly affects reproducibility.

**W12 (Soundness — Equation 2 design choice unexplained).** In Equation 2, the ego agent fuses its own raw feature $f_\varepsilon$ with the *reconstructed* collaborative features $\tilde{f}_i$ (Stage-2 output) rather than the protocol representations $p_i$ directly. The Stage-2 reconstruction adds an additional learned transformation that introduces potential information loss and computational overhead. Why is reconstruction to the local space necessary before fusion? An ablation comparing direct protocol-space fusion versus reconstruction-then-fusion would be informative.



### Minor Issues

- **Dense writing (Lines 171–181, col. 2):** This passage compresses multiple distinct technical concepts — SPD covariance construction, mask-guided spatial modulation, and the causal intervention objective — into a single run-on sentence. These ideas are each non-trivial and deserve separate treatment. Breaking similar passages into smaller, clearly sequenced steps throughout Section 3.2.2 would significantly improve readability.
- **Table 2 (\#Params column):** The unit for the "\#Params (New Modality)" column is not specified. Based on context this appears to be millions of parameters (M), but this should be stated explicitly in the table caption.
- **Figure 2 (SCM reference):** A brief pointer to an introductory SCM reference for readers unfamiliar with this graphical notation would be helpful.

---

> ### Author Rebuttal · Authors · 2026-03-31
>
> Thank you for your thorough review, insightful comments. Please see below for our responses to your comments.
>
> > ### Q1
>
> We understand the concern regarding higher-order statistics and spatial activation patterns, and we acknowledge that residual higher-order confounding may remain after intervention. However, we argue that the semantic factors $S$ inherently include spatial activation patterns, which should be preserved in the modality-invariant protocol space.
>
> Moreover, experiments under large modality discrepancies demonstrate that CauseCollab effectively mitigates modality-specific confounding. We also appreciate the feedback on Fig. 2: solid edges denote direct influences, dashed edges denote indirect influences, and the MGI-SPD intervention is designed to attenuate all non-causal associations marked by the red arrows.
>
> > ### Q2
>
> In CauseCollab, the protocol modality is set to PointPillar (0.4, 0.4, 4). Directly using the protocol modality introduces inherent LiDAR feature bias. Our method explicitly accounts for this by performing interventions on $f_p$ to mitigate the default anchor bias in the protocol space, rather than simply replacing one type of noise with another.
>
> As noted in STAMP (Gao et al., ICLR 2025), the alignment effectiveness depends on the affinity between the protocol and heterogeneous modalities. We conducted supplementary experiments and found that the causal intervention in CauseCollab effectively reduces this asymmetry, yielding comparable collaborative performance whether LiDAR or Camera is used as the protocol modality.
> |Protocol Modality|ALL|$L_{\text{pp4}}$ + $C_{\text{Eff}}$|$C_{\text{Eff}}$ + $L_{\text{pp4}}$|
> |-|-|-|-|
> |Lidar(PointPillar)|0.978 / 0.975|0.971 / 0.966|0.867 / 0.842|
> |Camera(Lift-Splat-Shoot)|0.975 / 0.973|0.968 / 0.964|0.875 / 0.847|
>
> > ### Q3
>
> We appreciate you sincerely pointing out the insufficient description of these components.
>
> The Mask Generator applies the classification head from single-modality training to local features to produce a confidence map $M$, which is then normalized, smoothed with a 2D Gaussian kernel ($k=5$, $\sigma=1.0$), and thresholded at 0.01 to obtain a binary mask.
>
> > ### Q4
>
> CGDR adopts 4-head FiLM, whose outputs are concatenated and projected back to $C$ channels by a 1x1 convolution.
>
> The gating function $g(\cdot)$ is a 1x1 convolution followed by SiLU, and is distinct from the Mask Generator. We will unify the notation and provide additional details in the final version.
>
> > ### Q5
>
> We tune the hyperparameters via grid search under AP\@0.3. The model is relatively sensitive to $\lambda_{\text{con}}$, achieving the best performance at 2.0, while the other Stage-1 hyperparameters are assigned relatively small weights.
>
> We first clarify a notation error in Section 3.2.2 below: the positive and negative samples for intervention are derived from the standard protocol feature $f_p$, rather than $p_i$. We apologize for the confusion.
>
> - $L_{\text{ssim}}$: enforces semantic consistency between $f_p$ and $p_i$.
> - $L_{\text{ctx-ssim}} / L_{\text{ctx-cos}}$: applied to $c_i$ and downsampled $p_i^+$.
> - $L_{\text{contrast}}$: contrastive loss with $p_i$ as anchor, $p_i^+$ as positive, and $p_i^-$ as negative.
> - $L_{\text{task}}^{(1)}$: protocol-head detection loss, implemented as focal loss plus smooth $\mathcal{L}_1$ loss.
>
> > ### Q6
>
> NegoCollab(Shao et al., NeurIPS 2025) mitigates the domain gap by negotiating a common representation, whereas our method explicitly disentangles semantic factors from modal entanglement to learn a modality-agnostic protocol space. Experimental results demonstrate that our method outperforms NegoCollab in both general and novel-modality scenarios.
>
> | Method | ALL | $L_{\text{sd}}^{\text{new}} + C_{\text{Eff}}^{\text{new}}$ | #Params (MB) |
> |-|-|-|-|
> |CauseCollab|0.978/0.975|0.966/0.959|0.68|
> |NegoCollab|0.974/0.972|0.958/0.953|0.98|
>
> > ### Q7
>
> MGI-SPD is used during the Stage-1 training of the Unified Converter; it is not involved during inference.
>
> Although the SPD matrix operation has a complexity of $O(C^3)$, the training time per epoch remains comparable to that of STAMP and is substantially lower than that of NegoCollab.
>
> |Method|Training Time per Epoch|
> |-|-|
> |CauseCollab|40 min|
> |STAMP|42 min|
> |NegoCollab|66 min|
>
> > ### Q8
>
> Thank you for the suggestion. We replaced SCE+CGDR with a single-tower ConvNeXt of comparable depth.
>
> |Method|AP@30|AP@50|
> |-|-|-|
> |CauseCollab|0.978|0.975|
> |w/o SCE+CGDR|0.968|0.966|
> |w/o CGDR|0.972|0.969|
>
> Combined with the ablation results on CGDR, these findings highlight the importance of SCE and further demonstrate the necessity of jointly incorporating both SCE and CGDR.
>
> > ### Q9
>
> $\Phi_{\varepsilon}(\cdot)$ and $H_{\varepsilon}(\cdot)$ are optimized for local-modal bias; directly fusing $p_i$ drops AP\@0.3 from 0.978 to 0.685.
>
>
> We look forward to your reply and would be happy to provide any further information.

---

> > ### Author Rebuttal · Reviewer_RPt6 · 2026-04-04
> >
> > **Resolved concerns:** W2 (LiDAR-biased anchor — new camera-protocol experiments are convincing), W4 (CGDR architecture now specified: 4-head FiLM, 1×1 conv + SiLU gating), W5 (loss term arguments clarified, notation error acknowledged), W6 (NegoCollab comparison added — CauseCollab outperforms), W7 (MGI-SPD is training-only — this resolves the inference overhead concern), W9 (SCE+CGDR vs. single ConvNeXt ablation provided and convincing), W12 (direct protocol-space fusion ablation provided).
> >
> > **Partially resolved** — W3 (Mask generator): The clarification that the mask generator reuses the classification head from single-modality pre-training is appreciated. However, this reveals a non-trivial dependency that was absent from the paper: the mask generator requires a pre-trained, per-modality detection head H_m to produce confidence maps M. While single-modality encoder pre-training is standard in this field, the specific reuse of H_m's spatial confidence output as a foreground mask should be documented..
> >
> > **Unresolved** — W1 (Causal justification): This remains my primary concern. The authors' response acknowledges that "residual higher-order confounding may remain" but argues that spatial patterns are part of S and that empirical results are strong. This does not address the formal concern: the paper uses the language of structural causal inference (counterfactual intervention, SCM) without providing identification conditions or formal justification that the SPD whitening–coloring transform satisfies the necessary criterion. Empirical performance does not constitute a causal proof. Either a formal justification should be added, or the contribution should be reframed as causally-motivated metric learning — which would be a more accurate and still compelling description of the work.
> >
> > I also note that several critical implementation details were only revealed in the rebuttal. These details must be incorporated into the main paper. Given the number and significance of the clarifications required, a substantial rewrite is needed that goes beyond what the rebuttal period allows. A revision that addresses W1 and fully incorporates the rebuttal clarifications would bring this paper to acceptance.

---

> > > ### Author Response · Authors · 2026-04-06
> > >
> > > We thank the reviewer for the careful reading and insightful comments, which reflect the attention and consideration given to our work. Below, we provide clarifications regarding the remaining concerns.
> > >
> > > > ### W3
> > >
> > > The single-modality head is obtained during single-modality pre-training and is reused in Stage-1. This point will be documented in the revised version.
> > >
> > > > ### W1
> > >
> > > We understand your concern regarding the causal justification of our SCM.
> > >
> > > 1.As a two-stage conversion framework for heterogeneous collaboration, we first consider a simple collaboration setting from modality $A$ to modality $B$. Before entering the $B$ reconstructor, the feature $F_{A \to p}$ still carries residual $A$-modality bias introduced by the simple ConvNeXt-based conversion, as well as protocol-modality bias, which we formalize as modality confounders $N_A$ and $N_p$. These confounders interfere with the local semantic reconstruction of modality $B$, because the $B$ reconstructor in previous methods has never been exposed to heterogeneous residual biases from other modalities. Therefore, we need to obtain a modality-invariant protocol semantic factor $S$. This is exactly the motivation of our method and the causal justification intended by Figure. 2.
> > >
> > > This can be formally expressed as follows:
> > >
> > > $ProtocolSpace_{\mathrm{before}} = \mathrm{Feature}(F_{A\to p}, F_{B\to p}, \ldots, F_{m\to p}) = \mathrm{Factor}(S, N_p, N_A, N_B, \ldots, N_m)$
> > >
> > >
> > > During Stage-1 training, by performing $do(F_p)$, we construct the positive sample $p_i^+$ (removing $N_p$) and the negative sample $p_i^-$ (away from injected $N_m$), such that only the semantic factor $S$ is retained.
> > >
> > > $ProtocolSpace_{\mathrm{after}} = \mathrm{Factor}(S)$
> > >
> > > Accordingly, during Stage-2 training, the modality-invariant protocol-space feature $F_{m \to p}$ (containing only $S$) is taken as input, so that local reconstruction is not affected by $N_m$ during both training and collaborative inference.
> > >
> > > 2.For the SPD intervention, Universal Style Transfer via Feature Transforms (Li et al., NeurIPS 2017) shows that first- and second-order statistics can serve as representations of domain bias in deep feature space. Since our protocol-space BEV representation is also a deep feature representation, we adopt the same representation-level view and model the dominant effects of $N_m$ and $N_p$ through the channel-wise first-order mean and second-order covariance of BEV features.
> > >
> > > Any clarification above will be included in the revised version.
> > >
> > > Finally, we again thank the reviewer for your thoughtful comments and insightful suggestions. Your concerns and suggestions will greatly help improve our work.

---

### Decision · Program_Chairs · 2026-04-30

**Decision:**

Accept (regular)

**Comment:**

This paper introduces CauseCollab, a two-stage framework for heterogeneous collaborative perception that utilizes a context-guided unified converter and a causally-motivated Mask-Guided Intervention (MGI-SPD) module to explicitly disentangle shared semantic factors from modality-specific statistical confounders. Initially, the reviewing committee praised the well-motivated problem framing, the parameter-efficient adaptation design for integrating new modalities, and the strong empirical gains demonstrated on both simulated (OPV2V) and real-world (DAIR-V2X) datasets. However, several critical critiques were raised, primarily concerning the formal theoretical justification of the causal claims—with reviewers questioning whether the method acts more as feature-level regularization than strict structural causal modeling—as well as missing architectural details, potential LiDAR bias in the protocol anchor, and a lack of hyperparameter sensitivity analyses. During the rebuttal phase, the authors diligently addressed these gaps by supplying comprehensive architectural specifics for the mask generator and CGDR, clarifying the training-only overhead of the MGI-SPD module, and providing new camera-protocol experiments, single ConvNeXt ablations, and baseline comparisons against NegoCollab. While one reviewer maintained lingering theoretical reservations regarding the strict formal proof of the causal framing, the committee ultimately agreed that the authors' extensive empirical evidence, thorough structural clarifications, and robust cross-modal performance successfully resolved the core practical and technical concerns, justifying a consensus of Weak Accept.